# DiscoX: Benchmarking Discourse-Level Translation task in Expert Domains

**Xiying Zhao**[1][*] **Zhoufutu Wen**[1][*] **Zhixuan Chen**[1], **Jingzhe Ding**[1], **Jianpeng Jiao**[1],
**Shuai Li**[1], **Xi Li**[1], **Danni Liang**[1], **Shengda Long**[1,2], **Qianqian Liu**[1], **Xianbo Wu**[1],
**Hongwan Gao**[1], **Xiang Gao**[1], **Liang Hu**[1], **Jiashuo Liu**[1], **Mengyun Liu**[1], **Weiran Shi**[1],
**Chenghao Yang**[1], **Qianyu Yang**[1], **Xuanliang Zhang**[1], **Ge Zhang**[1], **Wenhao Huang**[1]

[1]ByteDance [2]Peking University
{zhaoxiying, liniuniu}@bytedance.com

## Abstract

The evaluation of discourse-level translation in expert domains remains inadequate, despite its centrality to knowledge dissemination and cross-lingual scholarly communication. While these translations demand discourse-level coherence and strict terminological precision, current evaluation methods predominantly focus on segment-level accuracy and fluency. To address this limitation, we introduce DiscoX, a new benchmark for discourse-level and expert-level Chinese-English translation. It comprises 200 professionally-curated texts from 7 domains, with an average length exceeding 1700 tokens. To evaluate performance on DiscoX, we also develop Metric-S, a reference-free system that provides fine-grained automatic assessments across accuracy, fluency, and appropriateness. Metric-S demonstrates strong consistency with human judgments, significantly outperforming existing metrics. Our experiments reveal a remarkable performance gap: even the most advanced LLMs still trail human experts on these tasks. This finding validates the difficulty of DiscoX and underscores the challenges that remain in achieving professional-grade machine translation. The proposed benchmark and evaluation system provide a robust framework for more rigorous evaluation, facilitating future advancements in LLM-based translation. Our data and code are available at https://github.com/ByteDance-Seed/DiscoX

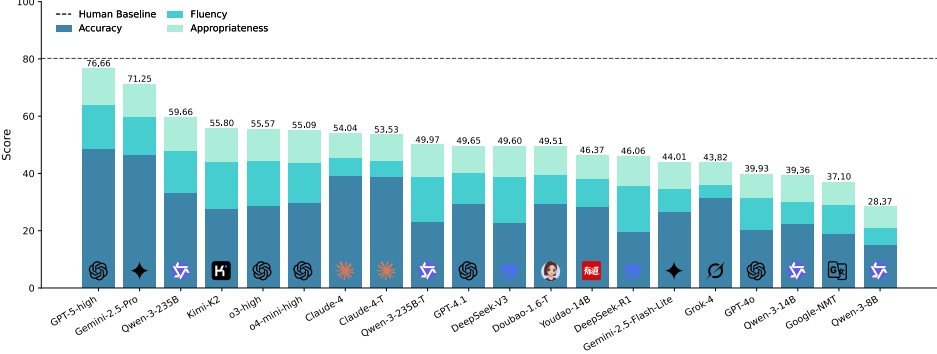

Figure 1: Leaderboard of DiscoX. This chart compares the overall translation performance of LLMs against a human expert baseline, which refers to the score of a translation produced by an expert human translator. Each bar is segmented to show the score composition from three dimensions: Accuracy (blue), Fluency (light blue), and Appropriateness (green).

---

[*]Equal contribution and Corresponding authors.

# 1 INTRODUCTION

Translation, a critical application of intelligent systems, is central to enabling cross-lingual communication and knowledge access. Recent advances in large language models (LLMs) have yielded substantial progress in segment-level translation, with state-of-the-art (SOTA) systems approaching human performance (Kocmi et al., 2024). However, notable shortcomings persist in the context of longer and more specialized texts (Wang et al., 2024b). Yet, existing benchmarks, such as WMT(Wang et al., 2023b), FLORES(Goyal et al., 2022) and Redtrans Bench(Guo et al., 2025b) predominantly focus on segment-level tasks,which means evaluating one or several sentences at a time. Consequently, they fail to assess whether models can sustain discourse-level coherence, handle domain-intensive terminology, or meet expert stylistic standards. This gap underscores the need for benchmarks designed to evaluate these advanced capabilities.

Such capabilities are critical in expert domains such as scientific articles, legal contracts, and technical manuals. For instance, as shown in Figure 2, in scientific articles, failing to sustain coherence across sections can distort the logical flow of arguments; in legal contracts, inconsistent translation of specialized terminology may weaken their binding force (Briva-Iglesias et al., 2024) and in technical manuals, imprecise or stylistically inappropriate renderings may lead to misunderstandings that jeopardize operational safety.

To address this gap, we introduce **DiscoX**, the first benchmark to evaluate the translation at the discourse-level and expert-level translation between Chinese and English (shown in Table 1). The benchmark, which cost 1,330 person-hours to create, is designed to simulate professional, real-world scenarios, comprising 200 cases across 7 domains, spanning both academic and non-academic contexts, with an average length of 1712 tokens. The construction of the dataset followed a multi-stage expert curation process. This process ensures that the test cases reflect authentic professional demands and incorporate domain-critical aspects such as terminology and cultural expressions. For each case, the key challenges are systematically collected and organized into expert-authored rubrics, such as the handling of ambiguous terminology. For example, in academic contexts, the abbreviation *LLM* may refer to *Large Language Model* in natural language processing, but to *Master of Laws* in the legal domain, highlighting the importance of precise terminology handling in context.

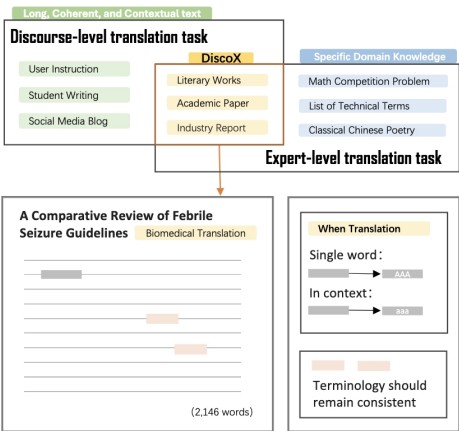

Figure 2: Definition of DiscoX. It illustrates the definition of two core concepts in DiscoX benchmark: "discourse-level translation" and "expert-level translation".

However, evaluating performance on a benchmark like DiscoX presents its own set of challenges, as conventional reference-based metrics are inadequate for long-form text and single-judge LLM evaluations can be unreliable (shown in Table 1) (Szymanski et al., 2025). To overcome this, we developed Metric-S, a novel, automated evaluation system based on the LLM-as-a-judge paradigm. Metric-S orchestrates multiple LLM agents in a structured workflow that includes pre-checking, quality estimation across accuracy, fluency, and appropriateness, error deduplication and attribution, and severity weighting. This modular framework exhibits strong alignment with human judgments (70.3% consistency on DiscoX) and significantly outperforms existing excellent reference-free met-

rics like XCOMET-QE (34.7%) (Guerreiro et al., 2024) on DiscoX. Together, DiscoX and Metric-S provide a principled and robust foundation for assessing professional-grade translation quality.

Table 1: Comparison of DiscoX with existing translation benchmarks. DiscoX distinguishes itself by (a) targeting discourse-level texts with a larger average length and focusing on expert domains. And, (b) its companion metric, Metric-S, offers reference-free and explainable evaluation, a unique feature among the compared methods.

**(a) Benchmark Comparison**

| Benchmark | Scenarios | Avg. Tokens |
|---|---|---|
| FLORES | General | 48.88 |
| WMT (2024) | News, Speech, Social, Lit. | 45.84 |
| Redtrans Bench | Social Conversation | 59.46 |
| **DiscoX (Ours)** | Academic & Non-academic | **1712.17** |

**(b) Metric Comparison**

| Metric | Ref-based | Explainability |
|---|---|---|
| N-grams | Yes | No |
| Neural Metric | Optional | No |
| LLM-as-a-judge | Optional | Optional |
| **Metric-S (Ours)** | **No** | **Yes** |

We apply DiscoX to a broad set of LLMs to test discourse-level and expert-level translation. As shown in Figure 1, even the strongest LLM (GPT-5-high) still lags behind professional human translators. This gap demonstrates that DiscoX serves as a realistic and challenging stress test for professional translation. Performance also diverges across dimensions: some models excel in accuracy, others in fluency, but none achieve a balanced, human-level competence. Overall, current models fall short of expert-level and discourse-level translation, underscoring both the difficulty of DiscoX and the need for future progress. Our contributions are:

- We present DiscoX, a benchmark that rigorously evaluates LLMs on discourse-level and expert-level translation tasks.

- We design Metric-S, a workflow-based automatic evaluation system for DiscoX, which enables more accurate and comprehensive assessments tailored to the requirements of professional translation domains.

- We conduct an extensive empirical study across multiple models, leveraging the explainability of Metric-S to reveal their respective strengths and limitations on challenging translation tasks.

- We discuss implications for system development and propose concrete recommendations to advance translation evaluation practices.

## 2 DISCOX

This section details the construction and composition of **DiscoX**. The benchmark is designed around two core concepts: (1) Discourse-level translation, which requires rendering a complete text as a single coherent unit, ensuring consistency in logic and style; and (2) Expert-level translation, which addresses highly specialized fields where the primary challenge is accurately conveying complex concepts and terminology, demanding subject-matter expertise. Based on these principles, we design a three-stage construction pipeline (Figure 3) to source texts from two primary and seven secondary domains. We next detail the expert team, construction process, and dataset composition.

### 2.1 DATA CONSTRUCTION

The construction of DiscoX is a large-scale collaborative effort involving 133 professionals (115 vertical domain experts and 18 linguistic specialists; see Appendix B for detailed profiles). The process is structured into three stages (illustrated in Figure 3).

**Data Annotation.** In the first stage, Vertical Domain Experts collect texts from their respective fields. Each source text has to meet three core requirements: (1) reflect authentic professional scenarios, (2) exceed a minimum length of 1,500 characters (Chinese) or words (English), and (3) be specific, self-contained, and amenable to the creation of unambiguous rubrics. Each text is then paired with a comprehensive set of expert-authored rubrics that delineate specific, verifiable evaluation criteria, including Grammar, Topic Terms, Terminology, and Culture-loaded Words. For

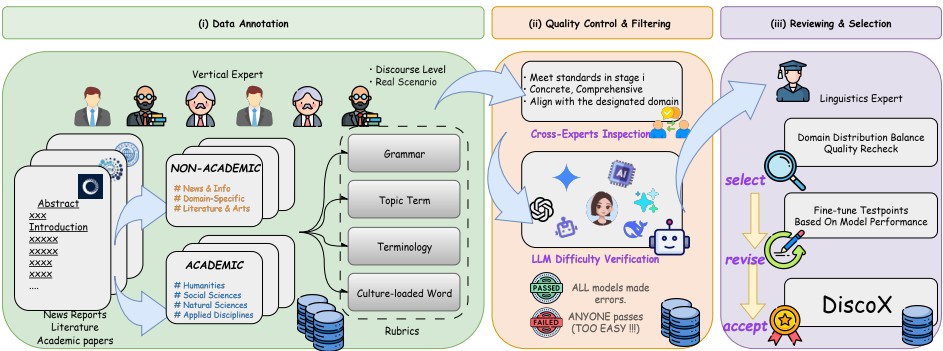

Figure 3: Overview of the rigorous curation pipeline for the DiscoX benchmark. This process transforms real-world texts into a validated evaluation set by employing a synergistic system of expert judgment and automated difficulty filtering.

instance, a rubric for a literary text: *"Checkpoint 1: The term ('yuanzi') in context must be translated as 'Ditan Park' or 'the park', not 'garden'."* This initial phase yields 665 potential tasks, with an average of 9.38 rubrics each. Detailed cases of rubrics can be found in Appendix C.3.

**Quality Controlling and Filtering.** The initial pool of 665 tasks then undergoes a rigorous quality control and filtering stage. This stage begins with a peer review by linguistic specialists to ensure textual professionalism. To establish a high and standardized difficulty, each task is subsequently tested against two SOTA LLMs (see Appendix C.1 for prompt structure). A task advances to the next stage only if both models fail on a minimum of eight predefined rubrics, signifying a stringent difficulty threshold. Specifically, a task is deemed sufficiently challenging only if both tested SOTA models fail to correctly translate content related to a minimum of eight predefined rubrics.

**Reviewing and Selection.** In the final stage, domain experts review the tasks that pass the difficulty filter. From this pool, they select the final 200 tasks, representing a selection rate of approximately 30%, ensuring a balanced and diverse distribution across domains. The experts then perform a final refinement, correcting any remaining flaws in the source texts and honing the rubrics based on the error patterns that are observed in the LLM outputs from the filtering stage. This step serves to maximize the benchmark's precision and evaluative utility.

## 2.2 DISCOX COMPOSITION

The resulting **DiscoX** benchmark comprises 200 high-quality translation tasks. As detailed in Table 2, the dataset is balanced across two primary domains: Academic (121 tasks) and Non-Academic (79 tasks), which are further subdivided into seven secondary domains. It spans the language pairs of English-Chinese (en→zh) and Chinese-English (zh→en). Notably, with an overall average input length of 1712.17 tokens, DiscoX texts are substantially longer than those in typical segment-level translation benchmarks.

## 3 METRIC-S: AUTOMATIC EVALUATION SYSTEM FOR DISCOX

Evaluating the discourse-level translations in DiscoX demands metrics that capture nuances beyond segment-level accuracy (Jiang et al., 2023; Zerva et al., 2024), such as fluency and appropriateness, which conventional automated metrics often fail to address (Wang et al., 2023b). We developed Metric-S, an automated evaluation system that leverages a multi-agent LLM system. As illustrated in Figure 4, it employs multiple LLMs to assess content in three dimensions: accuracy, fluency, and appropriateness. It then applies a hierarchical de-duplication process to attribute errors to their root causes before calculating a final, severity-weighted score. The prompt of different LLM judges is shown in Appendix C.2.

Table 2: Composition of the DiscoX benchmark. The table shows the breakdown of the 200 tasks into Academic and Non-Academic domains, further subdivided into seven specific fields. It presents the task count for each category and the average token length.

| Primary Domain | Secondary Domain | Count | Average tokens |
|---|---|---|---|
| Academic Papers | Social Sciences | 38 | |
| | Natural Sciences | 35 | 1875.58 |
| | Humanities | 28 | |
| | Applied Disciplines | 20 | |
| Non-Academic Tasks | News and Information | 37 | |
| | Domain-Specific Scenarios | 28 | 1450.49 |
| | Literature and Arts | 14 | |
| **Overall** | | **200** | **1712.17** |

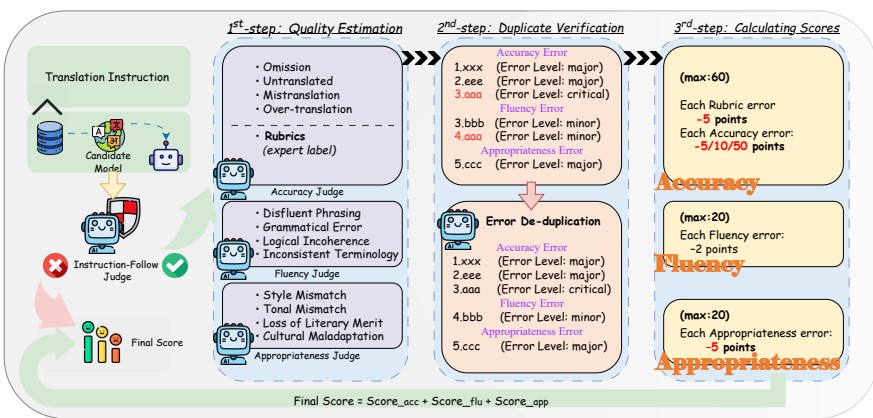

Figure 4: Overview of the Metric-S automated evaluation workflow. The system first employs an instruction-following judge to filter out invalid outputs. It then evaluate the translation's Accuracy, Fluency, and Appropriateness. Identified errors undergo a hierarchical de-duplication process to isolate root causes before a final score is computed based on the number and severity of the unique errors.

## 3.1 INSTRUCTION FOLLOWING CHECK

LLMs exhibit a tendency to deviate from translation instructions in discourse-level tasks, often defaulting to text continuation or summarization. To mitigate this issue, we employ an instruction-following check. Any output that fails to constitute a valid translation is immediately assigned a zero score and excluded from further evaluation.

## 3.2 QUALITY ESTIMATION

For outputs that pass the initial check, the system proceeds to evaluate quality across three dimensions: **accuracy, fluency, appropriateness**.

**Accuracy** The accuracy dimension evaluates how faithfully the translation preserves the source text's meaning, factual information, and emotional tone, while identifying issues such as mistranslation, omission, over-translation, or code-switching. Given that discourse includes professional articles across diverse domains, this metric also introduces the *rubrics* completed in the data annotation stage, enabling predefined specification of key terms, such as proper nouns and domain-specific terminology, that the model must handle correctly to receive credit.

**Fluency** The fluency dimension focuses on the quality of the translation from the perspective of the target language. The translated text is expected to be evaluated as if read by a native speaker,

with attention to linguistic smoothness, lexical consistency, and overall logical coherence. This requirement highlights a critical difference between discourse-level evaluation and segment-level evaluation, as fluency in extended discourse cannot be fully captured by traditional automatic metrics. It relies on LLMs' extensive linguistic resources and their capacity for self-assessment through learned analytic abilities.

**Appropriateness** The appropriateness dimension reflects a higher-level expectation of translation quality. Beyond basic usability, this metric seeks to discover the upper boundary of LLMs' translation capabilities. In addition to accuracy and fluency, appropriateness assesses whether culturally loaded expressions are properly rendered, whether the stylistic features of the source text are preserved, and whether the emotional tone and literary flavor are faithfully maintained in the translation.

## 3.3 Error deduplication and Attribution

In our multi-dimensional evaluation system, a single root error can propagate into multiple derivative issues. To prevent double-counting, Metric-S employs a hierarchical de-duplication and attribution process. This process isolates the fundamental cause of each error, ensuring that a single mistake is penalized only once. Specifically, errors marked as "Extremely Critical" in Accuracy take overriding priority, rubric-defined violations are systematically attributed to Accuracy, and for other overlaps, causal analysis determines which error is primary. *For example, if a lexical choice error leads to disfluent phrasing, only the Accuracy error is retained while the Fluency symptom is discarded.* Detailed example of de-duplication can be found at Appendix C.4.

## 3.4 DiscoX Score Calculation

The final score is defined as Score $= S_{\text{Acc}} + S_{\text{Flu}} + S_{\text{App}}$, where for each dimension $x \in \{\text{Acc}, \text{Flu}, \text{App}\}$, $S_x = MAX_x - \sum_{i=1}^{N_x} w_i^x e_i^x$. Accuracy, Fluency, and Appropriateness are weighted at 60, 20, and 20 points, respectively, which correspond to the maximum scores $MAX_x$. For each error, deductions are applied based on severity: 2 points for *minor*, 5 points for *major*, 10 points for *critical*, and 50 points for *extremely critical*. The details of severity levels in different domains can be found at E.

# 4 Experiments

## 4.1 Main Results

**Evaluation Setting.** In this section, we present the main results of DiscoX, covering 20 systems: 7 open-source LLMs, 11 closed-source LLMs, 1 domain-specific LLMs, and 1 neural machine translation (NMT) system. Gemini-2.5-Pro is further employed as the judge model within the Metric-S workflow[1]. The list of all models can be found at Appendix F.

**Results Analysis.** From the results in Table 3, we have the following observations. Discourse-level and expert-level translation remain a formidable challenge, and while general-purpose LLMs significantly outperform traditional MT systems, their performance still falls short of human standards. The results reveal a clear hierarchy, with the top model, GPT-5-high, achieving an overall score of 76.66 on the strength of its accuracy, yet still trailing human experts (80.16). Furthermore, performance is imbalanced across different evaluation dimensions, indicating distinct and complementary strengths. For instance, while GPT-5 excels in accuracy, other models like Kimi-K2 lead in fluency and appropriateness, and Claude-4 variants are highly accurate but struggle with fluency, indicating distinct and complementary strengths among the models. Specified cases of model performance on different dimensions can be found in Appendix D.2.1.

---

[1] Dec 2025 update: The result of using detailed instruction prompt and employing Gemini-3-Pro as the judge is shown in Appendix J.

Table 3: A ranked comparison of model performance on the DiscoX benchmark. The results highlight that even the most advanced models still trail the human expert. The data reveals imbalanced performance profiles, with different models excelling in distinct dimensions.

| Models | Overall | Accuracy | Fluency | Appropriateness | Open-source |
|---|---|---|---|---|---|
| Human Expert | 80.16 | 49.80 | 15.96 | 14.40 | - |
| GPT-5-high | **76.66** | **48.65** | 15.21 | **12.80** | ✗ |
| Gemini-2.5-Pro | 71.25 | 46.68 | 13.14 | 11.43 | ✗ |
| Qwen-3-235B | 59.66 | 33.15 | 14.96 | 11.55 | ✓ |
| Kimi-K2 | 55.80 | 27.63 | **16.44** | 11.73 | ✓ |
| o3-high | 55.57 | 28.78 | 15.79 | 11.00 | ✗ |
| o4-mini-high | 55.09 | 29.55 | 14.29 | 11.25 | ✗ |
| Claude-4 | 54.04 | 39.38 | 5.98 | 8.68 | ✗ |
| Claude-4-T | 53.53 | 38.98 | 5.47 | 9.08 | ✗ |
| Qwen-3-235B-T | 49.97 | 23.20 | 15.54 | 11.23 | ✓ |
| GPT-4.1 | 49.65 | 29.25 | 11.05 | 9.35 | ✗ |
| DeepSeek-V3 | 49.60 | 22.80 | 16.20 | 10.60 | ✓ |
| Doubao-1.6-T | 49.51 | 29.30 | 10.11 | 10.10 | ✗ |
| DeepSeek-R1 | 46.06 | 19.75 | 16.11 | 10.20 | ✓ |
| Gemini-2.5-Flash-Lite | 44.01 | 26.70 | 7.91 | 9.40 | ✗ |
| Grok-4 | 43.82 | 31.38 | 4.71 | 7.73 | ✗ |
| GPT-4o | 39.93 | 20.35 | 11.28 | 8.30 | ✗ |
| Qwen-3-14B | 39.36 | 22.40 | 7.73 | 9.23 | ✓ |
| Qwen-3-8B | 28.37 | 15.13 | 5.84 | 7.40 | ✓ |
| Youdao-14B | 46.37 | 28.50 | 9.82 | 8.05 | - |
| Google-NMT | 37.10 | 18.96 | 10.12 | 8.02 | - |

## 4.2 EFFECTIVENESS OF METRIC-S EVALUATION SYSTEM

To validate the effectiveness and reliability of Metric-S, we measured its pairwise consistency (Freitag et al., 2024) with professional human judgments. The experiment is designed to demonstrate Metric-S's superiority, particularly on our proposed benchmark, DiscoX.

**Evaluation Setting.** The evaluation is conducted on two test sets: our DiscoX benchmark and the WMT 2024 general translation task (Kocmi et al., 2024). We compared Metric-S against two high-performing baselines from the WMT 2024 metric shared task (Freitag et al., 2024): XCOMET-QE (Guerreiro et al., 2024) and ChrF (Popović, 2015). For each dataset, we randomly sampled 50 cases. In our evaluation, translations are scored by linguistic experts, and both human and metric scores are normalized to a $[0, 1]$ range for fair comparison. Since ChrF requires reference, we only evaluate its performance on WMT 2024 tasks.

To measure alignment with human judgments, we adopted the unified pairwise ranking consistency framework (Freitag et al., 2024). This method evaluates how often our metric agrees with human experts on which of two system outputs is superior. To ensure a robust evaluation, we made specific adjustments for different levels: at the system level, we used Soft Pairwise Accuracy (SPA) to provide a more nuanced comparison by accounting for statistical uncertainty in human rankings. To handle ties in the segment level, we treat metric scores as consistent if their difference is less than 0.05. Details of alignment framework can be found in Appendix G.

**Results Analysis.** The experimental results, summarized in Table 4, clearly demonstrate the superiority of Metric-S, particularly on our proposed DiscoX benchmark. The most striking finding is in the overall average consistency: Metric-S achieves 70.3%, more than doubling the 34.7% score of XCOMET-QE, a SOTA baseline. This significant gap highlights the failure of traditional metrics to handle complex, discourse-level phenomena. The divergence is especially pronounced at the system level, where XCOMET-QE's consistency plummets to just 10.0% on the zh→en task, while Metric-S maintains a robust 80.0%. These findings confirm that Metric-S provides a much more reliable

Table 4: Pairwise consistency of Metric-S and XCOMET-QE with human judgments. The table presents a comparison of evaluation metrics at both the system and segment levels for the DiscoX benchmark. ChrF is excluded because it requires a reference.

| Metric | Overall Avg. | System | | Segment | |
|---|---|---|---|---|---|
| | | zh→en | en→zh | zh→en | en→zh |
| Metric-S | **70.3%** | 80.0% | 90.0% | 54.8% | 56.4% |
| XCOMET-QE | 34.7% | 10.0% | 70.0% | 26.4% | 32.4% |
| ChrF | - | - | - | - | - |

and faithful alignment with human judgments for discourse-level translation evaluation. Case of the output of different evaluation methods can be found at Appendix D.1.

Furthermore, we investigated the internal validity of Metric-S to confirm that its advantage stems from the synergistic contribution of its core design components. Our ablation studies (detailed in Appendix A) reveal that while the full framework achieves 90% system-level consistency, removing the error de-duplication mechanism reduces this to 80%. More drastically, a simplified approach using a single LLM judge yields a mere 20% consistency. This evidence underscores that the effectiveness and stability of Metric-S are rooted in its integral, multi-component design.

## 5 ANALYSIS AND DISCUSSION

In this section, we move beyond raw scores to analyze what drives the observed performance differences. By investigating different perspectives, we seek to better understand both the current capabilities and the remaining limitations of LLMs in discourse-level translation.

### 5.1 LLM BETTER AT CHINESE-TO-ENGLISH THAN ENGLISH-TO-CHINESE

Our evaluation in Figure 5, detailed in Appendix H, reveals a significant performance asymmetry between translation directions, with LLMs consistently achieving higher scores in zh→en than in en→zh tasks. This gap is particularly pronounced for models like DeepSeek-V3, which shows a 34.8-point performance difference. In contrast, Doubao-1.6-T is the most balanced model, with a gap of only 7.2 points. This disparity is primarily driven by lower accuracy in en→zh translation, which we attribute to three main factors: (1) a data imbalance, where high-quality English corpora are more abundant than Chinese ones; (2) the prevalence of English-centric model architectures that require greater adaptation for generating Chinese; and (3) the inherent linguistic complexities of Chinese, such as its rich morphemes, implicit logic, and strong reliance on context and word order. Detailed cases showing performance asymmetry are provided in Appendix D.2.2.

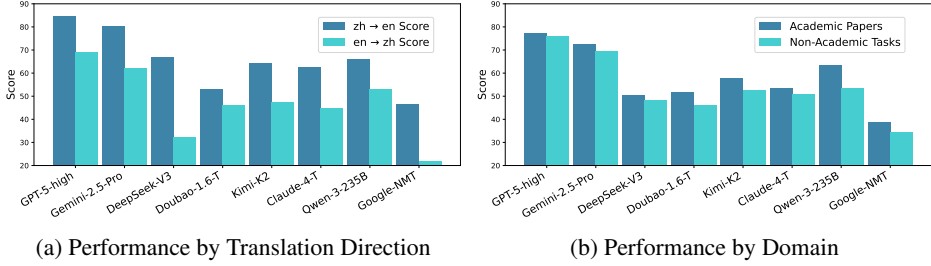

(a) Performance by Translation Direction  (b) Performance by Domain

Figure 5: Subplot (a) compares performance across translation directions, showing that models are stronger when translating into English. Subplot (b) compares performance across text domains, showing a clear advantage in translating academic papers.

## 5.2 STRONGER IN ACADEMIC PAPERS, WEAKER IN LITERATURE

Model performance exhibits a clear domain-based disparity, with significantly higher scores in academic translation compared to literary translation (illustrated in Figure 5, with detailed results in Appendix I). This gap is attributable to the inherent differences between the domains. Academic papers, with their structured and logical format, are more amenable to accurate machine translation. In contrast, literary works require models to interpret complex syntax, rhetorical devices, and cultural nuances—abilities that are difficult to evaluate and represent a key challenge for current LLMs. At the individual model level, GPT-5-high demonstrates strong generalizability by leading in both domains, whereas Kimi-K2 displays a particular strength in literary translation, reflecting its fluency-focused design.

## 5.3 PERFORMANCE DEFICIT OF THINKING MODELS

As shown in Table 5, Contrary to expectations, our results indicate that thinking-enhanced models consistently underperform their non-thinking counterparts in translation tasks. This performance deficit is particularly stark for Qwen-3-235B, where the non-thinking version scores nearly 10 points higher (59.66 vs. 49.97), a gap driven primarily by a significant drop in accuracy. A similar, though smaller, trend is observed with Claude-4. Error analysis suggests this underperformance is due to the thinking models' propensity to either over-summarize the source text, leading to omissions (under-translation), or introduce extraneous structural content (over-translation), thereby compromising translation faithfulness. Specified cases can be found in Appendix D.2.3.

Table 5: Comparison of Thinking vs. Non-thinking Models. The data shows that Non-thinking versions generally outperform their Thinking counterparts in translation tasks.

| Models | Type | Score | Accuracy | Fluency | Appropriateness |
|---|---|---|---|---|---|
| Qwen-3-235B | Thinking | 49.97 | 23.20 | 15.54 | 11.23 |
| | Non-thinking | 59.66 | 33.15 | 14.96 | 11.55 |
| Claude-4 | Thinking | 53.53 | 38.98 | 5.47 | 9.08 |
| | Non-thinking | 54.03 | 39.38 | 5.98 | 8.68 |

## 5.4 LIMITATIONS OF TRADITIONAL MT AND DOMAIN-SPECIFIC LLMS

Our results indicate that general-purpose LLMs significantly outperform traditional MT models and domain-specific LLMs in discourse-level translation tasks, whose performance is hindered by two primary factors.

**Input Length Constraints.** These models struggle with the discourse-level texts in DiscoX due to strict character limits. Even when document translation features are available, their output quality is often inferior, and some models produce disordered content like summarization or omissions on long inputs.

**Inferior Accuracy Compared to LLMs.** Compared to LLMs, these models exhibit lower accuracy, primarily due to omissions and mistranslations. They particularly struggle to correctly process domain-specific terminology and modern internet neologisms, a weakness that persists even in domain-specialized systems.

## 6 RELATED WORK

### 6.1 EVOLUTION OF MACHINE TRANSLATION BENCHMARKS

Traditional machine translation benchmarks such as WMT Wang et al. (2024a; 2023b) and IWSLT Agostinelli et al. (2025) have long served as the standard for evaluating translation quality. WMT, in particular, provides annually released test sets covering multiple language pairs and domains, but its focus has primarily been on sentence- or paragraph-level translation tasks. Similarly, IWSLT is specialized in speech and TED talk translations, offering relatively short and well-

structured content. While these benchmarks have played a crucial role in advancing MT research, they remain limited in both domain diversity and task complexity, leaving a gap in assessing translation performance under more challenging, real-world conditions.

In recent years, with the rapid improvement of models, translation benchmarks have increasingly emphasized broader coverage, higher task difficulty, and more diverse application scenarios. The FLORES-101 benchmarkGoyal et al. (2022) covers 101 languages with professionally translated sentences from diverse domains, enabling rigorous many-to-many multilingual translation evaluation. TransBenchLi et al. (2025) introduces a large-scale benchmark of 17k expert-curated tasks across 33 language pairs and multiple e-commerce scenarios, designed to evaluate machine translation systems at an industrial scale with greater domain coverage and linguistic difficulty. Redtrans BenchGuo et al. (2025b) introduces a benchmark of 2,858 zh→en and en→zh test cases covering informal, culturally loaded, and humor-rich SNS content that demands nuanced, context-aware translation beyond conventional MT settings. However, despite their contributions, these benchmarks mainly remain confined to sentence- or paragraph-level tasks, without extending to discourse-form translation or highly specialized domain-specific scenarios that demand expert-level knowledge and in-depth contextual reasoning. For instance, the WMT 2023 shared task on discourse-level literary translationWang et al. (2024a) highlighted this issue, as the low accuracy of traditional metrics led to results with questionable credibility.

## 6.2 Evaluation Metrics on Translation Tasks

Traditional machine translation metrics, including rule-based ones (e.g., n-gram overlap) and deep learning–based ones (e.g. COMET Rei et al. (2020), Metric-XJuraska et al. (2024)), achieve strong performance on sentence or paragraph level tasks because translation quality in this setting largely depends on local adequacy and fluency, which can be reliably captured by these methods. However, their evaluation accuracy declines when dealing with discourse-level texts, where evaluation requires modeling discourse-level coherence, consistency, and stylistic continuity—dimensions that sentence-level overlap or embedding similarity fail to representWang et al. (2023a). This mismatch between metric design and the global properties of discourse-level translation leads to weaker correlation with human judgmentsKocmi et al. (2024). Their reliance on reference translations also makes them poorly suited for discourse-level evaluation, where no single reference can capture the full range of valid outputsKim et al. (2024). In such cases, scores become unreliable, and in the absence of references, these methods often degrade sharply or cannot be applied at all. Compared with traditional metrics, LLM-as-Judge methods offer greater flexibility and interpretabilityDeutsch et al. (2025), as they can directly evaluate translations without strictly depending on reference outputs. However, in discourse-level scenarios, the capabilities of a single LLM judge may be constrained, leading to potential shortcomings in terms of comprehensive coverage across evaluation dimensions as well as accuracy. Evaluation using a single, general-purpose LLM is fraught with issues, including hallucinationsJi et al. (2023) and various biasesWang et al. (2023c). To better adjust general-purpose LLMs for single-model evaluation tasks, some researchers have begun to train smaller and more specialized evaluation models. These models are fine-tuned with extensive datasets featuring domain-specific evaluation criteria and illustrative examplesZheng et al. (2023).

## 7 Conclusion

We present DiscoX the first benchmark for assessing discourse-level, expert-level LLM translation, alongside Metric-S, a new reference-free evaluation system that aligns with human judgment. Our experiments show that while current LLMs are promising, they still underperform expert translators, struggling with global coherence, domain-specific terminology, and professional style. We are releasing both DiscoX and Metric-S to the community to foster research and advance the development of professional-grade translation models.

ETHICS STATEMENT

All data used in this study are collected from publicly available sources and do not contain any private or personally identifiable information. For data annotation, we engage qualified experts and provided fair compensation in accordance with market rates. We believe these practices ensure ethical compliance in terms of data usage, privacy protection, and fair treatment of human contributors.

REPRODUCIBILITY STATEMENT

We have made significant efforts to ensure the reproducibility of our work. A GitHub repository is provided in the abstract, which contains the full dataset, and implementations of the proposed evaluation metrics. Section 3 details the evaluation methodology, and Section 4 describes the experimental details.

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

## A ABLATION AND AUXILIARY EXPERIMENTS ON DISCOX AND METRIC-S

### A.1 MULTIPLE RUNS ON DISCOX

To probe the stability of model outputs and the quality of Data，We further perform multiple independent runs (three trials) for each model. As shown in Table 6, the negligible performance difference across trials indicates that the model produces consistent high-quality outputs across multiple samples, suggesting both strong model robustness and reliable evaluation.

Table 6: Scores of different models across three runs with mean and standard deviation.

| Model | First Score | Second Score | Third Score | Mean | Std. Dev. |
|---|---|---|---|---|---|
| Claude-4 | 52.73 | 53.21 | 53.52 | 53.15 | 0.40 |
| Claude-4-T | 53.28 | 54.03 | 54.33 | 53.88 | 0.54 |
| Gemini-2.5-pro | 71.53 | 72.51 | 73.47 | 72.50 | 0.97 |
| GPT-5 | 76.66 | 77.02 | 77.03 | 76.90 | 0.21 |
| Qwen3-235B | 57.10 | 59.70 | 59.00 | 58.60 | 1.36 |
| Qwen3-235B-T | 50.00 | 48.00 | 47.90 | 48.63 | 1.16 |

### A.2 ABLATION EXPERIMENTS ON METRIC-S

To validate the robustness and necessities of every elements in Metric-S, we carry out a series of ablation experiments.

#### A.2.1 EFFECTIVENESS OF THE WORKFLOW OF METRIC-S SYSTEM

We carry out the following experiments to illustrate Metric-S's effectiveness and necessities of elements in Metric-S system:

- Effectiveness of error deduplication mechanism: We evaluate the performance when the result of different judges is direclty applied to claculate final score without error deduplication.

- Metric-S vs single LLM judge: We try to merge all the prompts of 3 dimensions into one and employ a single LLM to judge model performances on DiscoX.

- Effectiveness of different dimensions and their weights: We also analyse the results when only the accuracy dimension is taken into consideration and when all 3 dimension exists but they are in the same weights.

We compare alignment with human judgments across Metric-S and all experimental settings. As shown in Table 7, Metric-S system achieves the best performance among all settings, demonstrating the integrity of the Metric-S framework and the necessity of its individual components.

Table 7: Pairwise consistency with human judgments under different experimental evaluation settings. The "Average" column reports the mean of system-level and segment-level consistency scores. All the experiments employ Gemini-2.5-pro as the baseline judge model.

| Metric | Average | System-level | Segment-level |
|---|---|---|---|
| Metric-S (Original) | 70.30% | 85.00% | 55.60% |
| Metric-S (No-duplicate) | 66.00% | 80.00% | 52.00% |
| Single Accuracy judge | 62.50% | 70.00% | 55.00% |
| Single LLM (Detailed prompt) | 55.90% | 60.00% | 51.80% |
| Metric-S (Same weight for all dimensions) | 48.20% | 50.00% | 46.40% |
| Single LLM (Simple prompt) | 24.90% | 20.00% | 29.80% |

### A.2.2 EFFECTIVE OF METRIC-S ON DIFFERENT LLM JUDGES

To examine which model suits best as the judge model and whether the strong performance of Metric-S is highly dependent on the choice of the judge model, we replace Gemini-2.5-Pro with several alternative baseline models as judges and compare their consistency with human judgments. As shown in Table 8, Gemini-2.5-Pro achieves the best performance among all models, illustrating the rationality of selecting it as the judge model. Moreover, Although the absolute scores of Metric-S vary moderately across different judge models, in all settings, Metric-S consistently outperforms the baseline evaluation metric XCOMET-QE by a substantial margin.

These results indicate that although the absolute accuracy of Metric-S is influenced by the underlying capability of the judge model, the observed robustness is not specific to Gemini. Instead, the stability of Metric-S reflects properties that are inherent to its evaluation design, rather than being an artifact of a particular judge model.

Table 8: Effect of different LLM judges on the consistency between Metric-S and human judgments. The table reports system-level, segment-level, and overall (average) consistency scores.

| Judge Type | System-level | Segment-level | Average |
|---|---|---|---|
| Metric-S (Gemini-2.5-pro) | 85.0% | 56.0% | 70.3% |
| Metric-S (DeepSeek-R1) | 70.0% | 45.6% | 57.8% |
| Metric-S (o3-high) | 70.0% | 46.4% | 58.2% |
| XCOMET-QE | 40.0% | 29.4% | 34.7% |

### A.3 EFFECTIVENESS OF SELF-PREFERENCE BIAS FOR LLM JUDGE

To validate whether self-preference bias of models affects the reliability of Metric-S, we compare the results of judgments of the three models from the previous subsection against human judgments. As shown in Table 9, Gemini-2.5-pro does not exhibit a self-preference bias when compared with human judgments. In contrast, o3 demonstrates a pronounced bias toward its own outputs, ranking itself first as a judge even though human experts placed it only third. These findings further justify the choice of Gemini-2.5-pro as the evaluation judge.

Table 9: Evaluation results of different judges on DiscoX samples.

| Model | Human | Metric-S+Gemini | Metric-S+Deepseek-R1 | Metric-S+o3-high |
|---|---|---|---|---|
| Gemini-2.5-Pro | 61.35 | 69.66 | 64.70 | 74.96 |
| Claude-4 | 57.45 | 67.90 | 64.86 | 71.10 |
| o3-high | 51.70 | 62.96 | 62.50 | 77.54 |
| DeepSeek-R1 | 51.25 | 67.82 | 63.04 | 72.78 |

### A.4 PERFORMANCE OF METRIC-S ON MWT 2024 TASKS

To further validate the robustness of Metric-S, we conducted an additional experiment comparing its performance with other traditional evaluation metrics on the WMT 2024 task. Considering these tasks include reference answers, we selected XCOMET Guerreiro et al. (2024)and ChrF Popović (2015), both reference-based metrics, for comparison.

As shown in Table 10, our experimental results indicate that, even without a reference, Metric-S can achieve human evaluation agreement rates close to those of traditional reference-based metrics on translation tasks beyond DiscoX. This reveals that Metric-S, as a robust, reference-free evaluation metric, is capable of effectively assessing model performance across various tasks without reliance on reference translations. More importantly, our evaluation method not only provides scores but also identifies error types and their severity, offering clear, actionable feedback. This transparency helps the evaluated parties understand their strengths and weaknesses, providing a clear path for targeted improvements.

Table 10: Pairwise consistency with human judgments on the WMT 2024 en→zh general translation task. This table compares the performance of the reference-free Metric-S against reference-based metrics. The "Average" column shows the mean of the system-level and segment-level scores.

| Metric | Average | System-level | Segment-level |
|--------|---------|--------------|---------------|
| Metric-S | 72.30% | 90.00% | 54.60% |
| XCOMET | 68.80% | 80.00% | 57.60% |
| ChrF | 55.70% | 60.00% | 51.40% |

## A.5 ACCUARCY OF LLM JUDGE OUTPUT IN DIFFERENT DIMENSIONS

Beyond consistency with human judgments at the score level, we further examine whether LLM-based judges produce accurate assessments at finer-grained, dimension-specific levels. To this end, we select five representative judge models and analyze their evaluations on the three assessment dimensions for the 50 cases introduced in Section 4.3. For each case, linguistic experts manually verify whether the model-generated judgments—covering both error types and error severity—are correct for each dimension.

As shown in Table 11, the accuracy for all 3 dimensions exceeds 95%, illustrating that Metric-S not only achieves high agreement with human judgments overall, but also demonstrates strong reliability and precision in detailed, dimension-level evaluations.

Table 11: Accuracy of judgment on different dimensions of Metric-S. The number of total cases is $50 \times 5 = 250$.

| Evaluation Type | Number of Correct Cases | Correct Rate |
|-----------------|-------------------------|--------------|
| Accurate Judge | 246 | 98.40% |
| Fluency Judge | 238 | 95.20% |
| Appropriateness Judge | 241 | 96.40% |

## B ANNOTATOR BACKGROUND

The integrity of the DiscoX dataset is ensured by a 133-member curation team operating under a dual-expertise model (summarized in Table 12). The team integrated 115 Vertical Domain Experts responsible for sourcing texts and authoring initial rubrics. They were supported by 18 Linguistic Experts, a group of certified MTI graduates with 5–16 years of experience. This latter group is tasked with ensuring linguistic accuracy, refining the evaluation criteria, and exercising overall project oversight.

Table 12: Table overview of the annotator team provides a breakdown of 133 experts responsible for the dataset curation, detailing the composition, number, professional experience, and background of the Vertical Domain Experts and Linguistic Experts.

| Composition | Number of Experts | Avg. Years | Background |
|-------------|-------------------|------------|------------|
| Vertical Domain Experts | 115 | 4–10 | Experts in practical fields are primarily professionals with 4–10 years of experience, while those in academic disciplines are mainly Master's/PhD holders from top-tier Chinese universities. |
| Linguistic Experts | 18 | 5–16 | Composed of professional linguists and translation specialists, primarily composed of Master's graduates in Translation and Interpreting (MTI) with high-level language certifications (e.g., TEM-8, IELTS 8.0) and 5-16 years of professional experience. |
| **Overall** | **133** | **4-16** | Expert in specific domain or linguistic. |

# C  DETAILED PROMPT AND RUBRICS

## C.1  SYSTEM PROMPT

> **System Prompt**
>
> Translate from Source Language to Target Language.  Do not output content beyond the translation.

## C.2  JUDGE PROMPT

Below is the complete prompt provided to the judge model, including the main stages of Instruction Following Judge, Accuracy Judge, Fluency Judge, Appropriateness Judge, and De-duplication Judge.

### C.2.1  INSTRUCTION FOLLOWING JUDGE

> **Instruction Following Judge**
>
> 任务
> 请帮我看下学生回答是否执行全文翻译指令:
> 主需问题（影响核心使用价值）
>
> > 1. 指令遵循 - 严重偏差
> >    - 输入为空，无译文
> >    - 译文与源文本语种一致，未进行翻译，转向问答，总结或其他内容
> >    - 输出译文大篇幅漏译，经比较漏译大于 50%
>
> 注意
> 注意，请仔细确定回答中是否有完整译文出现，若模型回复先对原文内容总结再进行翻译，亦视作满足翻译主需。
>
> 另外，如果发现模型的回复存在中英夹杂的现象，但总体而言还是在执行翻译指令，译语文本占比 50% 以上，亦视作满足主需。
> 输出格式
> 【质量问题检测】
> 助手 A:
> 是否存在问题: 是 / 否
> 问题等级: 主需问题 / 无
> (若存在，列出具体问题)
> [问题类型] - 具体说明
>
> 输出要求
> 问题等级只写综合评判后的最终结果
> 严格按输出格式进行输出
>
> 参考示例
> 示例 **1**:
> Les moyennes quadratiques
>
> 助手 A:
> "Les moyennes quadratiques sont un type particulier de moyenne utilisée en mathématiques pour des ensembles de valeurs, notamment lorsque l'on s'intéresse à des quantités qui dépendent de carrés, comme l'énergie (en physique) ou la dispersion statistique."
>
> 是否存在问题: 是
> 问题等级: 主需问题
> 未进行翻译。

### C.2.2 Accuracy Judge, Fluency Judge and Appropriateness Judge

---

**PE of Checkpoints**

你是一位翻译专家，擅长考察译文翻译质量。

现在，我已经准备好了需要检查的考点列表【checkpoints】，请你参考原文【prompt】和译文【response】，按默认顺序核查译文是否翻译正确，并输出json数组格式的报告。

注意：考点给出的译文为参考译文/推荐译文，如response中译文与要求不完全一致，但仍然是精确的翻译，亦视为正确。如果不够精确或偏离考点要求，不可以视作正确。

示例输出(输出所有考点的分析，无论正误)：

```
[{ 问题序号：1,
 问题段落：段落号,
 问题分析：你的分析,
 判断结果：正确/错误},
{ 问题序号：2,
 问题段落：段落号,
 问题分析：你的分析,
 判断结果：正确/错误}]
```

以下是你本次的任务：

---

**Judge PE of Accuracy**

我有一个复杂的翻译评估任务，需要评价超长文译文的准确度，自然流畅度及风格一致性三方面的内容。现在我将郑重把第一环节评估【译文准确度】的任务交给你，这是任务的评估的基石，请你审慎对待。

你是一位翻译专家，现在你需要评估以下译文的准确度。请按段落分组，以严格的标准，逐词对比原文与学生译文结合上下文，考察译文在翻译中是否存在偏离原文内容的现象。由于语言自然流畅度与风格一致性有后续的裁判评价，你不需要关注这些方面的内容，避免交叉评估导致重复。请注意，译文的读者为专业译员，对译文的准确度要求极高，请谨慎评估。

评估重点考察以下几类问题：

1. 错译：译文与原文词句含义不一致，导致翻译错误。译文词义准确，但与上下文语义不通，亦视作错译。
2. 漏译：译文与原文相比，缺乏部分事实信息，或忽略部分情感表达，导致翻译遗漏，语义不准。译文自主总结原文内容，导致语义产生偏移，视作严重漏译。
3. 未译：译文直接保留原文内容，未进行语言翻译，为严重错误。
4. 增译：译文与原文相比，添加额外事实信息或情感表达，导致与原文内容不一致。
5. 其他译文准确度问题。

其中，我要额外强调一类特殊的问题。部分学生译文中会出现以下未译情况：

例1：
原文：今天天气很好。
译文：The 天气 is pretty good.

"天气"一词非专业名词、人名、学术术语和地名等需要保留原文的特殊词汇，译文保留未翻。如果你在评估中遇到此类问题，严肃对待，标明【未译】标签，严重程度判定为【非常严重】。
具体地，

---

例2：
原文：今天天气很好。

译文：The 天气 (weather) is pretty good.
例2中尽管给出天气的译文**weather**，但在翻译过程中原文在前，译文在括号内，认定为【未译】，问题严重程度判定为【非常严重】。
例3：
原文：Victor Hugo is a writer.

译文：雨果（Victor Hugo）是一位作家。
例3中，尽管括号内保留了原文，但译文已按翻译规范给出，不视作未译，为正确译文。

问题严重程度

1. 普通（**Major**）：词级别的错译，漏译，影响语句理解，不影响段落理解，可视为中等错误。

2. 严重（**Critical**）：句级别错译，漏译，或额外增加原文无关信息的增译，导致大篇幅段落的含义出现偏差。段落标题等关键信息的错误。

3. 非常严重（**Extremely Critical**）：原文本保留源语言未译；大篇幅的漏译或通篇为总结性文本。

注意，在进行问题严重程度分类时，请审慎打出【严重（Critical）】标签。请仔细思考是否符合要求，如确实符合，则可打标为【严重（Critical）】。【非常严重（Extremely Critical）】标签只适用于我提到的特殊保留未译，以及极其严重的漏译情况。
评估结果中如有错误重复出现，不重复计错。脚注，标点，换行等格式问题不需要考虑，仅考虑文本内容。

输出格式
如存在以上问题，请按以下json示例格式输出，不要输出除此以外的内容：

[{ 问题序号：由你生成，从1开始，
 问题段落：段落号，
 问题类型：错译/漏译/未译/增译/其他，
 问题分析：你的分析
 问题严重程度：普通/严重/非常严重 }]

如果不存在任何问题，输出：

[{ 问题严重程度：整体无问题}]

以下是你本次的任务：

---

**Judge PE of Fluency**

一、角色及任务
你是一位翻译质量评估专家，并且你非常善于评估译文的自然流畅度。我会给你prompt原文和response译文，你现在的任务是基于【prompt原文】和以下评估标准，逐段、逐句对【response译文】质量进行分析和评分。

二、评估标准**&**问题类型

1. 语言流畅度：
   - 译文必须表达通顺、自然流畅，不能出现逐词死译、硬译的问题。
   - 译文不能生搬硬套原文的句式结构、词语搭配等，不能出现语言晦涩、拗口、结构混乱的问题。
   - 译文应符合目标语的语言表达习惯，不能出现翻译腔的问题。
   - 你需要特别注意长句、难句的表达是否自然流畅。

2. 语法正确性：译文的词法、句法不能存在问题，不能出现病句。

3. 逻辑连贯性：译文要尊重原文的行文逻辑，比如原文逻辑是并行的，则译文则不能是递进的；或原文逻辑是因果关系，则译文不能省略或者弱化因果的部分。中文句子间不太关注逻辑衔接，对只需意会的表达容忍度较高；而英文表达要求用词准确、逻辑清晰、表意明确。

4. 词汇一致性：文章中关键词的译法应保持前后一致，特别是人名、地名、专业术语、特色文化词汇等。如Syracuse这个地名有"锡拉库萨""叙拉古"等译法，若同一篇章中对此译法不一致，则很可能被误认为这是两个不同的地方。

特别注意：你无需关注翻译错误、中英夹杂等其他问题，只需要根据以上评估标准里提到的4个维度进行分析和评分即可。

三、问题严重程度分级

1. 有问题：译文存在语言流畅度、语法正确性、逻辑连贯性和词汇一致性中的任一问题。

2. 无问题：译文不存在语言流畅度、语法正确性、逻辑连贯性和词汇一致性的问题，整体表达自然流畅、符合译入语习惯。

特别注意：你找出来的每个问题都需要对应一个问题类型和问题严重程度的标签，不可将几个问题合并成一个问题类型和严重程度标签进行输出。

四、输出格式
如存在以上问题，请按以下json示例格式输出，不要输出除此以外的内容：

[ {
    问题序号：由你生成，从1开始，
    问题段落：段落号，
    问题类型：语言流畅度/语法正确性/逻辑连贯性/词汇一致性，
    问题分析：你的分析
    问题严重程度：有问题
},
{
    问题序号：2，
    问题段落：段落号，
    问题类型：语言流畅度/语法正确性/逻辑连贯性/词汇一致性，
    问题分析：你的分析
    问题严重程度：有问题
} ]

如果不存在任何问题，输出：

[ {
    问题严重程度：整体无问题
} ]

以下是你本次的任务：

---

**Judge PE of Appropriateness**

你将接收到一个重要的评估任务，这个任务主要分为三个环节：翻译准确度评估、翻译流畅度评估和翻译风格一致度评估，你主要负责翻译风格一致度评估环节。
你是一位严格的文学评论家，你极度敏锐，擅长洞察文本在风格、情感色彩、文采和文化层面的细微差异。你的评论严谨、深刻且一针见血，擅长从风格、情感色彩、文采和文化适应几个视角对译文进行审视。
我会给你prompt原文和response译文，你现在的任务是基于【prompt原文】和以下评估标准，逐句仔细找出【response译文】与原文风格、情感色彩、文采和文化适应中不匹配的地方，并按序号给出不匹配原因、不匹配内容和等级。

请注意：

1. 仅关注以下4个【评分维度】内容，请不要关注code switch（中英夹杂）、术语一致性、行文是否流畅及翻译腔等问题，这些问题已在准确度/流畅度评估环节识别出来。

2. 每个错误均需输出判分结果，不要将问题合并到一起输出。

评分维度:

1. 风格：是否合原文的语体和语域，使得译文和原文风格一致，如原文是法律文本，则译文用词正式、严谨，符合法律文本的语体。

2. 情感色彩：情感色彩是否和原文匹配。如原文是快乐的，译文也需要表现相同的快乐的情感色彩。

3. 文采：原文文本类型是文学体裁，则要相应考察译文是否有文采，如译文是否在语言运用上达到了与原文同等的生动性、形象性、节奏感和艺术美感。

4. 文化适应：译文中的比喻、典故、俚语、双关语等文化负载词是否得到了妥善处理，可以让译入语读者了解原文所表达的含义。

问题严重程度：
有问题：译文虽然可用，但在风格上未能100%复现原文的精髓。同一问题反复出现，不重复计错。

输出格式
如存在以上问题，请按以下json数组示例格式输出，不要输出除此以外的内容:

[{ 问题序号：由你生成，从1开始，
    段落序号：段落号，
    问题类型：风格/情感色彩/文采/文化适应，
    问题分析：你的分析，
    问题严重程度：有问题}]

如果不存在任何问题，输出:

[{    问题严重程度：整体无问题}]

【示例】
**¡prompt¿**
The prevailing neuroscientific paradigm posits that cognitive flexibility is fundamentally contingent upon neuroplasticity—the brain's capacity to reorganize its neural pathways. This process is largely driven by synaptic pruning and long-term potentiation, mechanisms that dynamically adapt in response to novel environmental stimuli and learning experiences.

**¡response¿**
现在主流的神经科学看法是，人的思维够不够灵活，根本上要看"神经可塑性"，也就是大脑重新组织神经连接的能力。这事儿主要是通过"突触修剪"和"长时程增强"来实现的，它们会根据新的环境刺激和学习经验来动态调整。

**¡评估结果¿**
[ 问题序号: 1,
问题段落: 1,
问题类型: 风格,
问题分析:原文中使用 'prevailing neuroscientific paradigm posits'，是严谨、客观、高度专业的学术语体，用词精准正式。译文却采用了'现在主流的看法'等口语化、甚至略带闲聊感的词汇，消解了原文的学术性和专业感。，
问题严重程度: 有问题 ]

### C.2.3    DE-DUPLICATION JUDGE

---

**Judge PE of De-duplication**

我 有 一 个 翻 译 评 估 任 务 ， 我 找 来 四 个 评 估 专 家 从accuracy， fluency，appropriateness，checkpoints (given)四 个 维 度 来 独 立 评 估 学 生 译 文 ， 我 担 心 几个维度的评估结果会彼此重复，导致最后扣两次分数，希望你帮我找出重复判分的地方，可以吗?
我会给你各个维度的评估结果，请找出重复判分的地方后帮我对其进行正确归因。

其中，accuracy的【非常严重（Extremly Critical）】标签是最高优先级，无论是什么内容，和谁重复，都保留【非常严重（Extremly Critical）】。
除此之外，无论哪个维度和checkpoints重复，均归因为checkpoints，这是第二优先级。

再次提醒，你只需要对裁判的评估结果中彼此重复的部分进行正确归因。以下情况，均不需要进行审查:

1. style的错误1属于accuracy维度。无需纠错
2. checkpoints和fluency均评估同一问题，fluency认为有问题，checkpoints认为正确。无需纠错。

因为这两种情况不属于【同一问题重复扣分两次】的范畴，相信你能明白。

关于输出，输出不需要任何总结性和承接性话语，请直接按照以下格式输出。分析简明扼要，不要阐释太多。
示例:

```
[{
 问题序号：1，
 重复维度：【checkpoints考点3】与【fluency问题1】，
 问题分析：你的分析，
 判断结果：此问题属于checkpoints，【fluency问题1】应删除
},
{
 问题序号：2，
 重复维度：    ，
 问题分析：你的分析，
 正确归因：
}]
```

如果评估不存在任何重复问题，输出:

```
[{
 本次评估无重复问题
}]
```

---

## C.3 CASE OF RUBRICS

The following are sample rubrics for evaluating [a specific task, e.g., academic papers]. In our evaluation, we use rubrics with a similar structure but with adjustments to the details for other categories of text to better accommodate the unique characteristics of each genre.

**Case 1**

**Primary Domain:** Academic Papers    **Secondary Domain:** Natural Sciences

**Prompt (excerpt)**

1、乳腺癌组织学分型
目前，乳腺癌的病理学诊断已从形态学结合免疫组化发展为形态学-免疫组化分子生物学特征相结合。精准的组织学分型对患者的预后判断、治疗决策有重要指导作用。如大部分三阴性乳腺癌(triple-negative breast cancer，TNBC)恶性程度高预后差，但也有一些低度恶性的TNBC生物学行为相对惰性，如分泌性癌、低级别腺鳞疹、纤维瘤病样梭形细胞癌、经典型腺样性癌等。对这部分低度恶性的TNBC除非有病理学检查证实的淋巴结转移，否则无需给予全身治疗。某些组织学类型的准确区分需行免疫组织化学和(或)分子病理学检测后确定部分组织学类型的乳腺癌具有独特的分子生物学特征，例如分泌性癌常伴有ETV6-NTRK3基因重排、经典型腺样囊性癌常有MYB-NFIB重排、低级别黏液表皮样癌常有 CRTC1-MAML2重排、极性翻转的高细胞癌常伴有IDH2基因突变。
2、乳腺癌组织学分级
组织学分级是重要的预后因素。推荐采用Nottingham分级系统对浸润性乳腺癌进行组织学分级。......只计数明确的核分裂象，不计数核浓染和核碎屑。
3、乳腺癌的分期
......肿瘤大小的测量有多种方法，包括临床体检、影像学评估、病理大体测量和显微镜下测量。
......

**Rubrics (excerpt)**

- "组织学分型" 推荐译为 Histologic type 或 Histologic subtype，保持全文一致。
- "生物学行为相对惰性" 推荐译为 Biologically indolent，避免用 vague 形容词。
- "纤维瘤病样梭形细胞癌" 必须译为 Fibromatosis-like spindle cell carcinoma。
- "全身治疗" 必须译为 Systemic therapy，保持肿瘤学标准术语。
- "基因重排" 必须译为 Gene rearrangement，保持分子病理学常用表达。
- "组织学分级" 推荐译为 Histologic grade，注意与 Histological grading 区分。
- "核浓染和核碎屑" 推荐译为 Pyknosis and karyorrhexis 或 Apoptotic bodies。
- "病理大体测量" 推荐译为 Gross pathologic measurement，注意 pathologic 与 pathological 均可但全文需统一。

**Case 2**

**Primary Domain:** Non-academic Tasks

**Secondary Domain:** Domain-Specific Scenarios

| Prompt (excerpt) | Rubrics (excerpt) |
|---|---|
| Liability, Indemnification and Release
... charges and expenses suffered or incurred by it in connection with the termination due to the negligence, breach of duty or other default or wrongdoing of the defaulting party, its servants, employees,agents or contractors.
...The CJV shall indemnify Party B against any loss or damage directly or indirectly suffered by Party B as a result of the failure of the Products manufactured hereunder to comply with the Technical Data("Defective Products")or to comply with such laws or regulations; provided, however, that such indemnification shall not exceed the total of ex-factory sales price, costs of delivery and transportation and other costs associated with the recall of these Defective Products.
Each Party hereby indemnifies the other Party and undertakes to hold harmless and defend the other Party against any and all claims, suits, losses, damages, disbursements (including legal and management costs) arising out of any alleged or actual breach or failure to comply with the terms and conditions hereof including but not limited to any infringement of the other Party's intellectual property or other rights occurring as a result of the offending Party's fault, omission or activities in connection with the Project...
(i) promptly notifies Consultant of any third party claim subject to indemnification hereunder... | • "servants"应译为"服务人员、雇员、代理人或承包商",不宜漏译,应与"employees, agents, contractors"并列处理。

• "any loss or damage"并列时须译作"任何损失或损害",不可合并成"损失"。

• "provided, however, that…"建议译为"但前提是",不能译为"但该等赔偿不应超过……"。

• "shall not exceed…"建议直译为"不得超过……",避免"该等赔偿不应超过"式弱化表达。

• "hold harmless and defend"建议译为"使……免受损害,并为其进行抗辩",漏译"defend"判错。

• "any and all claims, suits, losses"建议译为"任何及所有索赔、诉讼、损失"

• "offending Party"优先译为"过错方/侵权方",不可译作"违约方"。

• "promptly notifies"应译为"及时通知"。 |

**Case 3**

**Primary Domain:** Academic Papers    **Secondary Domain:** Humanities

| Prompt (excerpt) | Rubrics (excerpt) |
| --- | --- |

**Prompt (excerpt)**

标题：从 "默照" 到 "看话"：论禅宗 机锋 语言的 "不立文字" 与 "以言遣言" 之辩证

摘要： 禅宗，作为佛教在中国最独特的显现，其核心法门宣称 "不立文字，教外别传"。然而吊诡的是，禅宗 公案 与语录中又充满了 机锋、棒喝等极具表现力的语言运用。本文旨在剖析这一看似矛盾的现象，探究禅宗如何通过一种特殊的语言策略——"以言遣言"—— 来实现其 "不立文字" 的终极关怀。文章首先将回溯 "不立文字" 的本意，阐明其并非废弃言语，而是警惕语言作为 "指月之指" 的局限性。其次，本文将重点分析中唐以后 "机锋" 语言的兴起，以及宋代 "看话禅" 的成熟，是如何将语言从描述工具转变为一种能主动击碎思维定势的实践方法。通过对 "公案" 的参究和对 "话头" 的苦心孤诣，修行者被逼入 "大疑情" 的绝境，最终在言语道断处 "打破漆桶"，豁然开悟。本文认为，禅宗的语言革命，恰恰体现了其 "见性成佛" 之宗旨，即真正的超越，并非发生在语言之外，而是在语言的尽头。

正文:

一、"不立文字" 的本怀：作为 "指月之指" 的语言

禅宗提出 "不立文字，直指人心"，这常被误解为对一切文字和经典的彻底否定。实际上，禅宗并非主张缄默主义，而是深刻洞悉到语言的工具性与局限性。任何言教，无论多么精妙，都只是指向月亮的手指（"指月之指"），而非月亮本身。众生易犯 "认指为月" 的错误，即执着于经文的字面含义，而忘记其所指向的、超越言诠的实相本身。......

二、机锋 与 公案：从对话到实践的语言革命

中唐以降，禅宗大师们不再满足于平实的说法，转而发展出一种极其犀利、动态的对话方式 ——"机锋"

**Rubrics (excerpt)**

- "机 锋"推 荐 译 为"sharp verbal repartee"。
- "不立文字"必须译为"no reliance on words and letters"或"no-dependence on words and letters"。
- "以 言 遣 言"推 荐 译为"using words to eliminate words"或 "using words to dispel words"。
- "指 月 之 指"推 荐 译为"the finger pointing at the moon"或；"a finger pointing at the moon"。
- "看话禅"必须译为"Word Contemplation Chan (Kanhua Chan)"。
- "公 案"必 须 译为"koan(s)"；"gongan (koans)"。
- "大 疑 情"必 须 译 为"the Great Doubt"或"great doubt"。
- "话头"必须译为"the critical phrase (Huatou)"。
- "绝 境"推 荐 译为"desperate state"。
- "打 破 漆 桶"必 须 译为"smashing the lacquer bucket"或"breaking the lacquer bucket"。

## C.4 CASE OF DE-DUPLICATION

A single root error may lead to multiple, unfairly penalized issues across different dimensions. To address this, we introduce the Hierarchical De-duplications. This system ensures each error is traced back to its single fundamental dimension and penalized only once, eliminating redundant scoring. A case is shown blow.

**First Judge**

**acc_first_judge**

问题序号: 1
问题段落: 2
问题类型: 漏译
问题分析: "Does Machiavelli hold up Borgia as the model prince?"译文遗漏了这句，使得上下文的逻辑衔接不够紧密。
问题严重程度: 普通

**flu_first_judge**

问题序号: 1
问题段落: 3
问题类型: 逻辑连贯性
问题分析: "Does Machiavelli hold up Borgia as the model prince?"译文将此句省略，改变了原文的修辞效果，使得行文缺少了疑问到解答的节奏感。
问题严重程度: 有问题

问题序号: 2
问题段落: 6
问题类型: 逻辑连贯性
问题分析: 原文的论述逻辑是：先提出人文主义者的观点，并引用一句话来佐证，然后再引出马基雅维里截然相反的观点。而译文在呈现了人文主义者的观点后，省略了佐证其观点的引文，打乱了原文逻辑结构。
问题严重程度: 有问题

**app_first_judge**

问题序号: 1
段落序号: 3
问题类型: 风格
问题分析: "Does Machiavelli hold up Borgia as the model prince?" 译文将此句完全省略，直接转为陈述句'一些读者最初把《君主论》视为...'，这种处理方式虽然内容上无误，但改变了原文的语体风格。
问题严重程度: 有问题

问题序号: 2
段落序号: 7
问题类型: 风格
问题分析: 原文中的 'Machiavelli reinforces the prince's need to be feared by stating:' 是一个典型的学术写作中的过渡句...这改变了该句在段落中的功能，从而打断了原文连贯论证，影响了行文的风格
问题严重程度: 有问题

**De-duplication**

问题序号: 2
重复维度: 【accuracy问题1】、【fluency问题1】与【style问题1】
问题分析: 三者均指出了译文遗漏了原文关键设问句"Does Machiavelli hold up Borgia as the model prince?"的问题。accuracy将其归为"漏译"，fluency将其归为"逻辑连贯性"问题，style则认为其损失了"修辞效果"。根本原因为内容遗漏。
判断结果: 此问题属于accuracy，【fluency问题1】与【style问题1】应删除

---

**Final Judge**

| acc_final_judge | flu_final_judge | app_final_judge |

**acc_final_judge**

问题序号: 1

问题段落: 2

问题类型: 漏译

问题分析: "Does Machiavelli hold up Borgia as the model prince?"译文遗漏了这句，使得上下文的逻辑衔接不够紧密。
问题严重程度: 普通

**flu_final_judge**

问题序号: 2

问题段落: 6

问题类型: 逻辑连贯性

问题分析: 原文的论述逻辑是：先提出人文主义者的观点，并引用一句话来佐证，然后再引出马基雅维里截然相反的观点。而译文在呈现了人文主义者的观点后，省略了佐证其观点的引文，打乱了原文先立后破、观点对比的清晰逻辑结构。
问题严重程度: 有问题

**app_final_judge**

问题序号: 2

段落序号: 7

问题类型: 风格

问题分析: 原文中的 'Machiavelli reinforces the prince's need to be feared by stating:' 是一个典型的学术写作中的过渡句...这改变了该句在段落中的功能，从而打断了原文连贯论证，影响了行文的风格
问题严重程度: 有问题

---

# D    CASE ANALYSIS

## D.1    CASE OF METRIC FIDELITY VALIDATION

The assessment indicates that for the evaluation of model performance, the Metric-S system demonstrates a significantly higher correlation with expert human judgments compared to other metrics. Illustrative examples from the evaluation are provided below.

Case 1 is shown below:

---

**Prompt**

This phase is called the preliminary design, since this design is still subject to change. Usually, a crude, fast analysis method is adequate. At this stage, the cost of the structure is estimated, loads and member properties are revised, and the design is checked for possible improvements. The changes are now incorporated in the structure, a more refined analysis is performed, and the member design is revised. This project is carried out to convergence, the rapidity of which will depend on the capability of the designer. It is clear that a variety of analysis methods, ranging from "quick and dirty to exact", are needed for design purposes. An efficient analyst must thus be in command of the rigorous methods of analysis, must be able to reduce these to shortcut methods by appropriate assumptions, and must be aware of available design and analysis aids, as well as simplification permitted by applicable building codes. An up-to-date analyst must likewise be versed in the basics of matrix structural analysis and its use in digital computers, as well as in the use of available analysis programs or software.

---

**Response**

……对假定构型与恒载进行分析，依据结果设计构件（初步设计阶段）。此阶段适用快速近似分析法，随后进行成本估算、荷载与构件参数修正，并优化设计方案。改进后重新分析并修正设计，循环迭代直至收敛——效率取决于设计师能力。可见设计过程需要"从快速粗略到精确"的多层次分析方法。

高效的分析师必须：精通严谨分析法；能通过合理假设建立简化方法；熟悉现行设计分析工具及规范允许的简化条例；掌握矩阵结构分析原理及其计算机实现技术；熟练运用现有分析程序与软件。唯有如此，方能游刃有余地应对复杂结构设计挑战。

---

**Human Judge**

**Score:** 0.675

**Output:**
漏译："此阶段适用快速..."前漏译"since this design is still subject to change"；
瑕疵漏译："掌握矩阵结构分析原理及其计算机实现技术"前漏译"An up-to-date analyst must likewise..."；
错误增译："唯有如此...设计挑战"增译原文中没有的内容，错误；

---

**Metric-S**

**Score:** 0.580

**Output:**
问题类型："漏译"；
问题分析：原文"since this design is still subject to change"被漏译，影响了对"初步设计"原因的理解...；
问题严重程度："严重"；
问题类型："错译"；
问题分析：原文区分了"efficient"和"up-to-date"两种分析师，译文将两者合并，混淆了概念...；
问题严重程度："普通"；
问题类型："增译"；
问题分析：译文末尾增加了原文没有的总结性话语...；
问题严重程度："普通"；

---

**X-COMET**

**Score:** 0.177    **Output:** None

---

Case 2 is shown below:

---

**Prompt**

帮我翻译成英语：世界上每四瓶番茄酱中，就有一瓶产自新疆，这既是新疆的贡献，也是新疆的自豪。新疆番茄为什么"红"？地理条件得天独厚 被誉为番茄的"黄金产地"从异域传入中国的番茄非常喜欢充足的日照和剧烈的温差，而这些，新疆统统满足。日照丰富是新疆黄金产区的一大特色，新疆全年日照时数可达2550到3300小时，在全国各省区中位居第一。天山雪水的灌溉，成为番茄品质的加分项，因为天山雪水富含多种矿物元素，为生长在这里的番茄赋予更多矿物质和微量元素。

---

---

**Response**

One in Every Four Ketchup Bottles Worldwide Comes from Xinjiang—A Pride and Contribution of the Region Why Is Xinjiang Tomato So "Red"? Unrivaled Geographic Conditions: The "Golden Land" for Tomatoes Tomatoes, originally introduced to China from abroad, thrive under abundant sunshine and dramatic temperature swings—conditions Xinjiang perfectly provides. With annual sunshine hours reaching 2,550 to 3,300, Xinjiang ranks first among all Chinese provinces and regions. The region's tomatoes are further enhanced by irrigation from Tianshan mountain snowmelt, rich in minerals and trace elements.

---

**Human Judge**

**Score:** 0.525

**Output:**
文化适应：Why Is Xinjiang Tomato So "Red"未体现"双关"；
瑕疵漏译："With annual sunshine hours reaching 2,550 to 3,300, Xinjiang ranks first among all Chinese provinces and regions."这句话漏译大量原文信息；
错误漏译："The region's tomatoes are further enhanced by irrigation from Tianshan mountain snowmelt, rich in minerals and trace elements."这句话漏译大量原文信息，错误

---

**Metric-S**

**Score:** 0.400

**Output:**
问题类型："漏译"；问题分析：原文第一句为"日照丰富是新疆黄金产区的一大特色，新疆全年日照时数可达2550到3300小时……"。译文直接从日照时数开始翻译，完全遗漏了前半句"日照丰富是新疆黄金产区的一大特色"，导致段落的主题句缺失。；问题严重程度："严重"；

问题类型："文化适应"；问题分析：原文"红"字为双关语，既指番茄的颜色，也指其广受欢迎的程度。不可直译为"red"，推荐译为"hot"来表达"受欢迎"的含义。译文将其直译为"Red"，虽然加上了引号以提示其特殊用法，但未能有效传达"受欢迎、当红"这层核心含义；判断结果："错误"

---

**X-COMET**

**Score:** 0.211   **Output:** None

---

## D.2 CASE ANALYSIS OF DIFFERENT LLM MODELS

### D.2.1 CASE ANALYSIS OF LLM PERFORMANCE VARIATIONS ACROSS DIFFERENT DIMENSION

The models' performance is not balanced across the three dimensions (Accuracy, Fluency and Appropriateness). For example, Claude-4 performs well on Accuracy but poorly on Fluency-its translations are semantically correct but not smooth or natural. Conversely, DeepSeek-V3 performs well on Fluency but poorly on Accuracy.

**Prompt**

"全世界一共研究出13个番茄种的基因组，我们现在掌握11个。"新疆农业科学院副院长、研究员、加工番茄生物育种创新团队学科带头人余庆辉说。据介绍，该研究利用第三代单分子实时测序技术、全基因组光学图谱技术和高通量染色体构象捕获测序技术对11个番茄种进行基因测序组装，组装出11个染色体水平高质量基因组，打开了番茄属遗传资源领域研究的宝库，使新疆从种植大区一跃成为世界番茄遗传资源富集区。

**Classification:** News and Information; zh-en

---

**GPT-5-high Response**

"Globally, the genomes of 13 tomato species have been characterized; we now have 11 of them," said Yu Qinghui, [...] a world-leading reservoir of tomato genetic resources.

**Score: 90**
accuracy: 60 — fluency: 20 — appropriateness: 10

**Analysis:** Accurate terminology, fluent prose, the translation exemplifies the professional style of science news.

---

**Claude-4 Response**

"Worldwide, 13 tomato species genomes have been **researched** in total, and we now **possess** 11 of them," said Yu Qinghui, [...] **opening up** a treasure trove of research in the field of tomato genetic resources, transforming Xinjiang from a major planting region to a world-rich area for tomato genetic resources.

**Score: 63**
accuracy: 45 — fluency: 8 — appropriateness: 10

**Analysis:** The meanings of "research" and "possess" are correct, but the contextual flow is not smooth; "opening up" is a literal translation and fails to capture the metaphorical meaning.

---

**DeepSeek-V3 Response**

"Out of the 13 tomato species genomes researched worldwide, we now have **mastered** 11," said Yu Qinghui, [...] elevating Xinjiang from a major planting region to a global hub for tomato genetic resources.

**Score: 46**
accuracy: 25 — fluency: 16 — appropriateness: 5

**Analysis:** "mastered" is a mistranslation and does not fit the original meaning.

### D.2.2 ASYMMETRY IN TRANSLATION DIRECTIONALITY: AN ENGLISH-TO-CHINESE CASE STUDY

The case reveals a significant performance disparity, with models performing considerably worse when translating from English to Chinese compared to the reverse direction. For instance, even the top-ranked GPT-5-high shows a noticeable quality gap between its English-to-Chinese output and the translations produced by professional human experts.

---

**Prompt**

Machiavelli further went on to question the loyalty of the citizens and advised the Prince that "...because men a wretched creatures who would not keep their word to you, you need not keep your word to them."

The same to Hanfeizi. ......  He believed that human beings were driven by the greed for profit. We can see some trace in his famous saying: strict mother has kind children, while kind mother has brute children.

Both of them believed that the relationship among human beings was a kind of naked interest‿oriented relationship.

---

**Classification:** Humanities; en-zh

---

**Reference Response (Human)**

马基雅维利进一步质疑民众的忠诚度，并告诫君主：“…… 因为人性是卑劣的，他们不会信守对你的诺言，所以你也不必守对他们的诺言。”
韩非子也持相同观点。…… 他认为，人是被追逐利益的贪欲所驱使的。从他的名言“严家无悍虏，而慈母有败子”中，便可窥见一斑。
他们二人都认为，人与人之间是一种赤裸裸的利益关系。

---

**GPT-5-high Response**

他进一步质疑臣民的忠诚，并告诫君主：“既然人是卑劣的生物，不会对你守信，你也不必对他们守信。”
韩非子的看法亦然。…… 他认为人受逐利之心驱动。这一点可从他那句名言“严母出孝子，慈母多败儿”中略见端倪。二人都认为，人际关系是一种赤裸裸的利益取向的关系。

---

**Score: 81**
accuracy: 45  —  fluency: 16  —  appropriateness: 20

---

**GPT-5-high Response Judge Analysis**

{问题序号: 2, 问题分析: 原文中的"strict mother has kind children, while kind mother has brute children"是对韩非子思想的英文转述。译文将其翻译为更为通俗的"严母出孝子，慈母多败儿"。考点要求翻译为更贴近《韩非子·显学》原文"慈母有败子，而严家无格虏"的"严家无悍虏，而慈母有败子"。虽然译文与考点推荐译文意思相近，但译文使用的是一个民间俗语，而考点要求的是更具经典出处的表述，因此译文不够精确，未满足考点要求。, 判断结果: 错误,}

{问题序号: 1, 问题类型: 语言流畅度, 问题分析: 原文开头为"The same to Hanfeizi."，译文为"韩非子的看法亦然"。此处的"亦然"虽然意思正确，但作为段落开头，与上文衔接略显生硬和书面化，不够自然。更通顺的表达可以是"韩非子也是如此"或"韩非子也持相同看法"。问题严重程度: 有问题}

### D.2.3 Case Analysis Between Thinking LLM and Non-Thinking LLM

The study finds that thinking models generally underperform in non-thinking translation tasks. Thinking models are more prone to omitting information or providing summarized translations, which may lead to information loss.

---

**Prompt**

Incidents may be diagnosed and resolved by people in many different groups, depending on the complexity of the issue or the incident type. All of these groups need to understand the incident management process, and how their contribution to this helps to manage the value, outcomes, costs, and risks of the services provided: Some incidents will be resolved by the users themselves, using self-help. Use of specific self-help records should be captured for use in measurement and improvement activities.

**Classification:** Academic Papers; en-zh

---

**Non-thinking Response**

根据问题或事件类型的复杂程度，事件可能由多个不同团队的人员进行诊断和解决。所有这些团队都需要理解事件管理流程，以及他们的参与如何帮助管理所提供服务的价值、成果、成本和风险：有些事件将由用户通过自助服务自行解决。应记录特定自助服务记录的使用情况，用于测量和改进活动。

**Non-thinking Score: 71**
Accuracy: 45 — Fluency: 6 — Appropriateness: 20

---

**Thinking Response**

根据事件复杂度或类型，诊断与解决可能涉及多个团队：用户可通过自助服务解决部分事件，相关记录应纳入测量与改进活动。

**Thinking Score: 14**
Accuracy: 0 — Fluency: 4 — Appropriateness: 10

---

## E   Severe Levels in Different evaluation dimensions of Metric-S

**Accuracy**. Representing the foundation of a translation, this dimension assesses factual correctness and usability. Errors are heavily penalized and stratified by their impact: *major* for word- or phrase-level mistakes, *critical* for paragraph-level deviations, and *extremely critical* for discourse-level failures like large-scale omissions.

**Fluency**. Evaluating the "goodness" of a translation, this dimension focuses on whether the text is well-formed and reads naturally. As fluency issues typically affect the reading experience rather than the core meaning, all such errors are uniformly classified as *minor*.

**Appropriateness**. Measuring the pursuit of "excellence"，this dimension assesses stylistic and cultural resonance. Errors in tone, style, or cultural adaptation can significantly alter a text's intended impact; thus, any such error is classified as *major*.

## F    MODELS EVALUATED ON DISCOX

- **Open-source LLMs (7):** Kimi-K2, DeepSeek-V3 (Liu et al., 2024), DeepSeek-R1 (Guo et al., 2025a), Qwen-3-235B-A22B-Instruct[2], Qwen-3-235B-A22B-Thinking[3], Qwen-3-14B, Qwen-3-8B (Qwen Team, 2025).
- **Closed-source LLMs (11):** GPT-4o, GPT-4.1, o3-high, o4-mini-high, GPT-5-high (Achiam et al., 2023; OpenAI, 2025b; 2024; 2025a), Claude-4-Sonnet-Thinking[4], Claude-4-Sonnet[5] (Anthropic, 2025), Doubao-1.6-Thinking[6], Gemini-2.5-Pro, Gemini-2.5-Flash-Lite (Google Cloud, 2025b;a), WebCrawl-grok4-common[7].
- **Domain-specific LLMs (1):** Youdao-14B [8].
- **NMT system (1):** Google Translate[9].

## G    CORRELATION FRAMEWORK OF EVALUATING ALIGNMENT OF DIFFERENT METRICS WITH HUMAN JUDGMENTS.

To evaluate alignment with human judgments, we adopt a unified *pairwise ranking consistency* framework (Freitag et al., 2024). For any two model outputs $a$ and $b$ under the same evaluation unit (system or segment), let $s^H(a), s^H(b)$ denote human scores and $s^M(a), s^M(b)$ denote metric scores. A pair is counted as consistent if

$$\text{sign}\big(s^H(a) - s^H(b)\big) = \text{sign}\big(s^M(a) - s^M(b)\big).$$

- At the system-level, we employ Soft Pairwise Accuracy (SPA). Unlike standard pairwise accuracy, SPA accounts for the statistical uncertainty in rankings, providing a more nuanced comparison of system performance without arbitrarily penalizing metrics for statistical ties.
- At the segment-level, we use the metric, a segment-level accuracy measure with tie calibration. If $s^H(a) = s^H(b)$, we additionally treat the pair as consistent when $|s^M(a) - s^M(b)| < 0.05$ during the segment-level calculation.

The final correlation score for each metric is the average of the results across all distinct tasks: system-level and segment-level for both en→zh and zh→en translation directions.

## H    DETAILED RESULTS OF MODEL PERFORMANCE ON CHINESE-TO-ENGLISH AND ENGLISH-TO-CHINESE TASKS

The detailed results of model performance on Chinese-to-English and English-to-Chinese tasks are shown in Table 13.

## I    DETAILED RESULTS OF MODEL PERFORMANCE ON DIFFERENT DOMAINS

The results of model performance on all primary and secondary domains is shown in Table 14.

## J    MODEL PERFORMANCE WHEN USING DETAILED INSTRUCTION PROMPTS AND EMPLOYING GEMINI-3-PRO AS THE JUDGE

During our experiment, we detect that with the simple translation instruction as the input, some models will produce somewhat unprofessional translations to align with preferences, hindering to

---

[2]abbr. Qwen-3-235B
[3]abbr. Qwen-3-235B-T
[4]abbr. Claude-4-T
[5]abbr. Claude-4
[6]abbr. Doubao-1.6-T
[7]abbr. Grok-4
[8]Input limits: 10,000 characters
[9]abbr. Google-NMT; 5,000 character input limit, with document upload support for longer text

Table 13: Comparison of model performance on zh→en and en→zh translation tasks. The table presents a detailed breakdown of scores for each model across the two translation directions. The 'Diff' column quantifies the performance gap between zh→en and en→zh translations.

| Models | zh→en | | | | en→zh | | | | Diff |
|---|---|---|---|---|---|---|---|---|---|
| | Score | Accuracy | Fluency | Appropriateness | Score | Accuracy | Fluency | Appropriateness | |
| GPT-5-high | 84.49 | 52.35 | 17.24 | 14.90 | 68.83 | 44.95 | 13.18 | 10.70 | 15.66 |
| Gemini-2.5-Pro | 80.22 | 50.25 | 15.82 | 14.15 | 62.26 | 43.10 | 10.46 | 8.70 | 17.96 |
| Qwen-3-235B | 66.15 | 36.35 | 16.80 | 13.00 | 53.17 | 29.95 | 13.12 | 10.10 | 12.98 |
| Kimi-K2 | 64.12 | 32.90 | 18.32 | 12.90 | 47.46 | 22.35 | 14.56 | 10.55 | 16.66 |
| o3-high | 67.18 | 36.10 | 17.98 | 13.10 | 43.95 | 21.45 | 13.60 | 8.90 | 23.23 |
| o4-mini-high | 70.34 | 40.10 | 15.94 | 14.30 | 39.84 | 19.00 | 12.64 | 8.20 | 30.50 |
| Claude-4 | 62.44 | 43.45 | 7.84 | 11.15 | 52.62 | 35.30 | 4.12 | 6.20 | 9.82 |
| Claude-4-T | 62.34 | 44.15 | 6.94 | 11.25 | 44.70 | 33.80 | 4.00 | 6.90 | 17.64 |
| Qwen-3-235b-T | 58.12 | 28.45 | 15.92 | 13.75 | 41.81 | 17.95 | 15.16 | 8.70 | 16.31 |
| GPT-4.1 | 65.82 | 39.65 | 13.62 | 12.55 | 33.48 | 18.85 | 8.48 | 6.15 | 32.34 |
| DeepSeek-V3 | 66.97 | 36.55 | 17.62 | 12.80 | 32.23 | 9.05 | 14.78 | 8.40 | 34.74 |
| Doubao-1.6-T | 53.13 | 33.65 | 9.18 | 10.30 | 45.89 | 24.95 | 11.04 | 9.90 | *7.24* |
| DeepSeek-R1 | 58.12 | 28.60 | 16.72 | 12.80 | 34.00 | 10.90 | 15.50 | 7.60 | 24.12 |
| Gemini-2.5-Flash-Lite | 62.51 | 38.75 | 11.26 | 12.50 | 25.51 | 14.65 | 4.56 | 6.30 | **37.00** |
| Grok-4 | 59.29 | 40.70 | 7.04 | 11.55 | 28.33 | 22.05 | 2.38 | 3.90 | 30.96 |
| GPT-4o | 58.13 | 30.95 | 15.88 | 11.30 | 21.73 | 9.75 | 6.68 | 5.30 | 36.40 |
| Qwen-3-14B | 47.20 | 26.80 | 9.00 | 11.40 | 31.51 | 18.00 | 6.46 | 7.05 | 15.69 |
| Google-NMT | 46.49 | 25.51 | 11.90 | 9.08 | 27.80 | 12.47 | 8.36 | 6.97 | 18.69 |
| Qwen-3-8B | 32.95 | 18.70 | 6.20 | 8.05 | 23.78 | 11.55 | 5.48 | 6.75 | 9.17 |
| Average | 61.37 | 36.00 | 13.22 | 12.15 | 39.94 | 22.11 | 9.71 | 7.75 | 21.43 |

evaluate their translations performances accurately. So we update a version of dataset with detailed translation instruction prompts to reduce the impact of preference alignment(data released in our Github codebase). Moreover, we also find that Gemini-3-Pro behaves well on multiple benchmarks, revealing its potential to be a good judge.

Taking all things into consideration, we add an experiment using detailed instruction prompt and Gemini-3-Pro as the judge. Notably, this experimental setting is consistent with that adopted in the Seed-1.8 technical report. The results are shown in Table 15.

# K USE OF LLMS

In this paper, we use LLM for polishing writing only. Specifically, the models are used to refine grammar, improve sentence fluency, and enhance the clarity and readability of the text, without altering the underlying meaning or introducing new content.

Table 14: Detailed model performance on academic versus non-academic tasks in DiscoX. The table provides a fine-grained breakdown of model rankings and scores across seven sub-domains, categorized under Academic and Non-Academic Texts. In the header, 'R' and 'S' stand for Rank and Score, respectively.

| Models | Academic Papers | | | | | | | | | | Non-academic Tasks | | | | | | | |
| | Overall | | Humanities | | Social Sciences | | Applied Disciplines | | Natural Sciences | | Overall | | Domain-Specific Scenarios | | Literature and Arts | | News and Information | |
| | Rank | Score | R | S | R | S | R | S | R | S | Rank | Score | R | S | R | S | R | S |
|---|---|---|---|---|---|---|---|---|---|---|---|---|---|---|---|---|---|---|
| GPT-5-high | 1 | 77.07 | 1 | 71.93 | 1 | 78.79 | 1 | 84.25 | 1 | 75.23 | 1 | 76.03 | 1 | 76.00 | 1 | 68.29 | 1 | 78.97 |
| Gemini-2.5-Pro | 2 | 72.32 | 2 | 67.39 | 2 | 75.63 | 2 | 76.35 | 2 | 70.37 | 2 | 69.58 | 2 | 69.32 | 2 | 64.07 | 2 | 71.86 |
| Qwen-3-235B | 3 | 63.58 | 3 | 58.86 | 3 | 66.18 | 3 | 69.70 | 3 | 61.03 | 5 | 53.66 | 3 | 51.00 | 5 | 43.00 | 6 | 59.70 |
| Kimi-K2 | 4 | 57.88 | 4 | 54.86 | 5 | 61.11 | 7 | 63.15 | 6 | 53.80 | 7 | 52.58 | 7 | 47.82 | 3 | 55.57 | 10 | 55.05 |
| o3-high | 5 | 56.02 | 7 | 52.11 | 4 | 64.29 | 6 | 63.70 | 13 | 45.80 | 3 | 54.86 | 4 | 50.00 | 4 | 49.00 | 5 | 60.76 |
| o4-mini-high | 6 | 55.36 | 5 | 53.50 | 6 | 58.45 | 4 | 64.25 | 10 | 48.40 | 4 | 54.68 | 8 | 47.43 | 7 | 40.86 | 3 | 65.41 |
| Claude-4 | 8 | 54.57 | 11 | 49.14 | 8 | 55.61 | 5 | 63.80 | 7 | 52.51 | 6 | 53.20 | 5 | 49.71 | 9 | 35.00 | 4 | 62.73 |
| Claude-4-T | 7 | 55.30 | 12 | 47.50 | 7 | 57.68 | 8 | 62.95 | 5 | 54.57 | 8 | 50.80 | 6 | 48.43 | 12 | 33.36 | 7 | 59.19 |
| Qwen-3-235B-T | 11 | 51.34 | 10 | 49.18 | 13 | 49.74 | 18 | 49.10 | 4 | 56.09 | 10 | 47.86 | 10 | 44.86 | 6 | 41.29 | 12 | 52.62 |
| GPT-4.1 | 9 | 52.07 | 6 | 52.18 | 10 | 52.84 | 12 | 57.40 | 11 | 48.11 | 12 | 45.94 | 14 | 37.82 | 11 | 33.50 | 8 | 56.78 |
| DeepSeek-V3 | 12 | 50.58 | 14 | 43.89 | 11 | 51.97 | 10 | 58.80 | 9 | 49.71 | 9 | 48.10 | 12 | 42.32 | 8 | 39.14 | 9 | 55.86 |
| Doubao-1.6-T | 10 | 51.67 | 8 | 50.43 | 9 | 54.39 | 17 | 51.40 | 8 | 49.86 | 11 | 46.20 | 9 | 46.93 | 14 | 32.21 | 14 | 50.95 |
| Youdao-14B | 13 | 49.98 | 13 | 45.12 | 16 | 47.00 | 9 | 62.50 | 12 | 46.79 | 15 | 40.72 | 13 | 41.58 | 15 | 28.50 | 18 | 44.79 |
| DeepSeek-R1 | 14 | 48.31 | 9 | 50.04 | 12 | 50.68 | 13 | 55.95 | 17 | 39.97 | 14 | 42.62 | 11 | 42.71 | 16 | 28.43 | 16 | 47.92 |
| Gemini-2.5-Flash-Lite | 15 | 46.24 | 15 | 39.18 | 14 | 49.58 | 11 | 58.50 | 16 | 41.26 | 16 | 40.59 | 17 | 31.43 | 18 | 21.93 | 11 | 54.59 |
| Grok-4 | 16 | 44.12 | 16 | 34.57 | 15 | 47.21 | 15 | 54.20 | 14 | 42.66 | 13 | 43.33 | 15 | 35.89 | 10 | 34.29 | 13 | 52.38 |
| GPT-4o | 18 | 39.60 | 17 | 33.25 | 17 | 43.97 | 16 | 52.60 | 20 | 32.51 | 17 | 40.43 | 16 | 32.11 | 13 | 33.07 | 15 | 49.51 |
| Qwen-3-14B | 17 | 42.68 | 19 | 32.68 | 17 | 43.97 | 14 | 54.70 | 15 | 42.40 | 19 | 34.27 | 19 | 28.86 | 17 | 22.64 | 19 | 42.76 |
| Google-NMT | 19 | 38.86 | 18 | 33.18 | 19 | 41.35 | 19 | 47.00 | 18 | 36.03 | 18 | 34.42 | 18 | 30.50 | 19 | 15.50 | 17 | 44.83 |
| Qwen-3-8B | 20 | 32.79 | 20 | 22.96 | 20 | 31.66 | 20 | 46.65 | 19 | 33.94 | 20 | 21.59 | 20 | 18.64 | 20 | 10.57 | 20 | 28.00 |

Table 15: A ranked comparison of model performance on the DiscoX benchmark using detailed instruction prompt and Gemini-3-Pro as the judge model.

| Models | Overall | Accuracy | Fluency | Appropriateness |
|---|---|---|---|---|
| Gemini-3-Pro | **75.83** | **48.75** | 17.82 | 8.88 |
| GPT-5.1-high | 75.06 | 45.85 | 18.71 | **10.50** |
| DeepSeek-R1 | 73.90 | 46.33 | 18.76 | 8.85 |
| Gemini-2.5-Pro | 73.48 | 46.15 | 18.08 | 9.25 |
| Claude-Sonnet-4.5 | 71.65 | 46.76 | 16.55 | 8.69 |
| Qwen3-Instruct | 65.70 | 37.10 | 18.47 | 10.18 |
| DeepSeek-V3.2 | 64.55 | 38.80 | 16.60 | 9.15 |
| GLM-4.6 | 63.90 | 39.40 | 16.59 | 7.88 |
| o3-high | 60.60 | 32.40 | **18.83** | 9.40 |
| o4-mini | 59.10 | 31.33 | 17.77 | 9.98 |
| Qwen3-T | 59.10 | 30.78 | 18.20 | 10.10 |
| Gemini-2.5-Flash-Lite | 56.00 | 32.18 | 15.75 | 8.06 |
| DeepSeek-V3 | 55.90 | 28.33 | 18.50 | 9.05 |
| Kimi-K2 | 52.07 | 24.53 | 18.55 | 9.18 |
| GPT-4o | 50.40 | 25.70 | 16.81 | 7.90 |

