# OpenReview forum: "DiscoX: Benchmarking Discourse-Level Translation in Expert Domains"
_ICLR.cc/2026/Conference — ICLR 2026 Poster_

### Official Review · Reviewer_y79T · 2025-10-30

**Soundness:** 3
**Presentation:** 3
**Contribution:** 3
**Rating:** 6
**Confidence:** 5

**Summary:**

This paper presents a benchmark and evaluation system, DiscoX, for discourse-level and expert-level translation tasks, particularly for Chinese-English translation. The system introduces Metric-S, a reference-free evaluation metric that uses large language models (LLMs) to score translations based on three key dimensions: accuracy, fluency, and appropriateness. The authors demonstrate that Metric-S aligns closely with human judgment, achieving significant improvements over traditional reference-based metrics. The paper also introduces a novel multi-stage task construction pipeline to create high-quality translation tasks for expert domains, addressing a significant gap in discourse-level machine translation evaluation.

**Strengths:**

1. The paper tackles the important issue of evaluating discourse-level translation, a topic often neglected in standard translation benchmarks that typically focus on segment-level accuracy. This approach is timely and relevant, especially given the increasing use of LLMs for complex, domain-specific translation tasks.
2. The authors propose a robust evaluation metric (Metric-S) that combines accuracy, fluency, and appropriateness in a reference-free manner. The empirical results show that the metric aligns closely with human evaluation, demonstrating the potential of LLMs for fine-grained translation assessments.
3. The multi-stage task creation pipeline for DiscoX ensures that the translation tasks reflect real-world professional demands. The quality control process, including peer reviews and filtering based on SOTA models, adds credibility to the dataset construction.
4. The proposed score-guided multi-round translation pipeline shows clear improvement in translation quality across various models, such as QWQ, GPT-5, and DeepSeek. This is a practical and useful contribution for real-world applications of machine translation.

**Weaknesses:**

1. While the Metric-S system is empirically sound, the paper does not provide a theoretical framework explaining why certain dimensions (accuracy, fluency, appropriateness) are weighted in the manner they are, or how these dimensions interact. The authors should consider adding an analysis of the sensitivity of these weights, or provide a more detailed justification for their choices.
2. Metric-S relies on a single evaluation model, and there is limited analysis of how different LLMs might perform as evaluators. To validate the robustness and independence of Metric-S, the authors should consider conducting experiments comparing different LLM models as judges and analyzing the consistency of their evaluations across models.
3. The current experiments focus on Chinese-English translation. To make the framework more generalizable, the authors should consider testing Metric-S on other language pairs (e.g., English-Spanish, English-German) or multilingual tasks to assess its robustness across languages and domains.
4. The paper mentions that a task is included in the DiscoX benchmark only if it passes a stringent threshold (at least 8 rubric failures from two SOTA models). However, the rationale behind this threshold is not explained. It would be helpful if the authors clarified the reasoning for this cutoff or conducted experiments showing how the difficulty threshold affects the overall task difficulty distribution.

**Questions:**

1. How robust is Metric-S across different domains, such as technical writing or legal texts, where semantic trade-offs may differ from general text translation?
2. Has Metric-S been tested on out-of-domain or noisy data (e.g., user-generated content) to check its stability and potential biases?
3. Would combining Metric-S with embedding-based metrics (e.g., BERTScore) improve the overall evaluation reliability, especially for more diverse domains?
4. How computationally intensive is the multi-round feedback pipeline, and is it scalable for use on large translation datasets?

---

> ### Author Response · Authors · 2025-11-25
>
> # Weakness1
> We thank the reviewer for this comment regarding the weighting of evaluation dimensions.
> Our weighting scheme is carefully designed based on professional translation assessment principles and is strongly supported by empirical evidence.
> 1. Rationale for the 60/20/20 Weighting:
> The prioritization of Accuracy (60%) reflects the paramount importance of factual fidelity and terminological precision in expert-level translation, where semantic integrity is the primary determinant of a translation's utility. While Fluency and Appropriateness (20% each) remain essential for readability and style, their secondary weighting aligns with established industry rubrics (e.g., MQM) and expert criteria from professional translation programs.
> 2. Empirical Sensitivity Analysis:
> To quantitatively validate our design, we conducted an ablation experiment using a uniform weighting scheme (1:1:1). The results show a substantial drop in agreement with human judgments, confirming the superiority of our original weights.
> | Method  | System-level Consistency | Segment-level Consistency | Total Consistency |
> |------------------------------------------|---------------------------|----------------------------|--------------------|
> | metric-s + Gemini (60/20/20)  | 0.85 | 0.556 | 0.703 |
> | metric-s + Gemini (with average weight)  | 0.50 | 0.464  | 0.482 |
> - System-Level Consistency: 0.85 → 0.50
> - Total Consistency: 0.703 → 0.482
> This significant performance degradation demonstrates that the 60/20/20 distribution is not arbitrary but is critically aligned with human evaluation priorities. We will include these results and the corresponding analysis in the revised manuscript.
> 3. Interaction among dimensions.
> As discussed in our response to Reviewer 2, Q2, the three dimensions in Metric-S (Accuracy, Fluency, Appropriateness) interact through the system’s core mechanisms—multi-agent judging, rubric-based constraints, and hierarchical deduplication—which explicitly separate root causes of discourse-level errors. Multi-agent judging ensures that each dimension captures a distinct class of long-range phenomena; rubric-based scoring enforces document-level constraints that prevent one dimension from implicitly absorbing another; and hierarchical deduplication isolates root causes so that symptoms across dimensions do not artificially inflate correlations. As a result, the dimensions provide complementary signals rather than redundant ones, supporting the need for a weighted combination rather than a uniform or single-dimension score.

---

> ### Author Response · Authors · 2025-11-25
>
> # Weakness 2
> Thank you for raising this point regarding the robustness of using single LLM as judges.
> To address the concern, we analyze the potential sources of bias in Metric-S from two complementary perspectives:
> (1) bias arising from the LLM-as-judge itself, and
> (2) bias arising from the Metric-S system design (multi-agent workflow, scoring structure, and error deduplication).
> We present experiments targeting each dimension separately to show that both the judge model and the system design are robust, and that the overall evaluation remains stable across different LLM evaluators.
> 1. Gemini-as-Judge evaluation  exhibits extremely low factual  judgement bias.
> Translation evaluation in our setting is objective: factual mistranslations, omissions, and additions can be directly checked against the source text.
> To quantify judge correctness independent of human preference, we manually reviewed 50 samples × 5 models × 3 dimensions, and each dimension achieved ≥95% factual correctness. This confirms that Gemini is not producing unstable or hallucinated judgments in this task.
> ## Human Evaluation Accuracy of Metric-S
> | Evaluation Type| Number of Correct Cases | Total Number of Evaluations | Correct Rate |
> |---|--|--|--|
> | Accurate judge of Metric-s         | 246 | 250 | 0.984|
> | Fluency judge of Metric-s          | 238  | 250  | 0.952  |
> | Appropriateness judge of Metric-s  | 241  | 250  | 0.964  |
> 2. Metric-S is robust across judge models.
> (1) Multi-agent robustness across independent judges. Metric-S is a multi-agent evaluation system. Even when replacing Gemini with independent judges, the overall trends remain stable:
>  ## Consistency Comparison (Different Judges)
> | Judge Type| System-level Consistency | Segment-level Consistency | Total Consistency |
> |--|--|--|--|
> | Metric-s (Gemini as judge)     | 0.85  | 0.56     | 0.703   |
> | Metric-s (Deepseek-R1 as judge)| 0.70    | 0.46   | 0.578     |
> | Metric-s (GPT-O3 as judge)     | 0.70       | 0.46    | 0.582   |
> | XCOMET-QE         | 0.40     | 0.29    | 0.347    |
>
> The consistently higher performance across judges demonstrates that the observed stability is not specific to Gemini, but inherent to the design of Metric-S.
>
> (2) Multi-agent robustness across independent judges. To test whether prompting alone could explain the results, we applied simple, detailed, and Metric-S to Gemini:
> ## Prompting Variants with Gemini as Judge
>
> | Method  | System-level Consistency | Segment-level Consistency | Total Consistency |
> |--|-----|--|--|
> | Metric-s + Gemini        | 0.85  | 0.56| 0.703|
> | Detailed prompt + Gemini | 0.60    | 0.52     | 0.559 |
> | Simple prompt + Gemini   | 0.20  | 0.30   | 0.249  |
> The Metric-S design (structured roles, multi-dimensional scoring, error deduplication) consistently outperforms single-prompt baselines, confirming that robustness arises from the system rather than prompt artifacts.
>
> In summary, Gemini-2.5-Pro, as the judge, delivers the highest accuracy and lowest bias in judge outputs. Together with the Metric-S system, it provides an evaluation solution that achieves the highest agreement rate with human assessment, fulfilling the experimental design objectives. In the future, we will continue to explore ways to further reduce evaluation bias in large language model judges.
> # Weakness 3
> We thank the reviewer for this constructive feedback.
> The primary objective of this study was to establish a robust evaluation framework for discourse-level and expert-level translation tasks.  Our findings show that the proposed metric-s system is highly reliable, showing superior correlation with human judgments.
> Inspired by the current benchmarks (PaperBench with 20 tasks and TauBench with 165 tasks) are widely used but relatively small, so we intentionally prioritized validating the methodology before scaling to multilingual settings. Having established the reliability of Metric-S—demonstrating strong agreement with human judgments—we are now expanding the benchmark in two directions:
> 1. Scale Enhancement: The test suite will be expanded to 500 texts, with a minimum of 50 instances per sub-category to ensure statistical power.
> 2. Linguistic Diversification: We are extending the language coverage to include Chinese-Japanese (ZH-JA), English-French (EN-FR), and English-German (EN-DE) pairs.
> In summary, the test bank will be expanded to at least 1,500 texts. This follow-up work is already in progress, and we welcome your continued interest.

---

> ### Author Response · Authors · 2025-11-25
>
> # Weakness 4
> Thank you for raising this question regarding the rationale behind the rubric count threshold.
> The choice of “8 failed rubrics” was not arbitrary; it resulted from an empirical analysis conducted during the pilot stage of DiscoX task construction. The full process is as follows.
> Here's a detailed explanation of the procedure:
> 1. Expert-curated pilot set (60 tasks): We began with a pool of 60 source texts and asked domain experts to exhaustively annotate all potential challenging rubrics, including specialized terminology, complex discourse structures, and nuanced expressions. These tasks typically contained 8–12 rubrics each, as detailed in Appendix B.3.
> 2. Stress-testing with SOTA models.: These 60 candidate tasks were then evaluated using five high-performing SOTA LLMs.  For each task, we counted the number of rubrics that multiple models consistently failed.
> 3. Empirical difficulty distribution. Among the 10 empirically hardest tasks (lowest average model performance):
>   - The number of failed rubrics ranged between 5 and 10.
>   - The mean was 7.4 failed rubrics per task.
>   This indicates that:
>   - Tasks with only 3–5 failed rubrics were too easy: they typically touched only one or two localized issues and did not exhibit discourse-level depth.
>   - Tasks with 10+ failed rubrics were too dense: models failed nearly every rubric, and Metric-S required significantly longer evaluation time (due to high error frequency; Section 3 workflows), making such tasks impractical for large-scale benchmarking.
> Based on this empirical distribution, we established a threshold of 8 failed rubrics. This ensures that the tasks included in the final DiscoX benchmark are sufficiently challenging, discriminative, and meaningful for evaluating the capabilities of advanced translation systems.
> We will clarify this data-driven selection process in the revised manuscript to enhance the reproducibility of our benchmark construction.
> # Question 1
> Thank you for this thoughtful observation.
> Metric-S is designed to be domain-agnostic, and our experiments confirm that it remains robust across diverse text types—even when the semantic trade-offs differ substantially, such as in technical, legal, journalistic, or humanities writing.
> To evaluate cross-domain robustness, we conducted a targeted analysis of human–metric consistency across four representative sub-categories in DiscoX (each covering distinct professional domains, including legal contracts, biomedical texts, technical manuals, academic prose, and news writing; see Table 2 in the paper
> Key Finding: Metric-S demonstrates consistently strong performance across diverse text types, with human alignment stability ranging from 0.67 to 0.79—significantly outperforming the XCOMET baseline, which achieved only 34.7% average consistency on the same benchmark.
> The detailed consistency rates are as follows:
> - Humanities: 0.77
> - Natural Sciences: 0.70
> - News & Information: 0.79
> - Domain-Specific Scenarios: 0.67
> These results demonstrate that Metric-S does not rely on domain-specific heuristics. Its multi-agent judging, rubric-based evaluation, and hierarchical deduplication mechanisms allow it to adapt to the different semantic priorities in each domain—whether factual precision in technical writing, terminological rigor in legal texts, or stylistic consistency in humanities literature.
> In summary, Metric-S maintains stable and high consistency across all tested domains, confirming its robustness for evaluating heterogeneous real-world translation tasks.
> # Question 2
> Thank you for raising this point regarding performance on diverse data types.
> To address this, we have evaluated Metric-S on the WMT 2024 en→zh general translation task—which contains web text, news reports, informal writing, and user-generated content—material that is substantially noisier and stylistically different from the long-form professional texts in DiscoX.
> As detailed in Appendix G.4, Metric-S achieved a 72.3% average consistency with human judgments on this dataset, surpassing the reference-based XCOMET (68.8%) and ChrF (55.7%). This demonstrates that our evaluation framework maintains high agreement with human assessment even on noisier, more heterogeneous text types.
> These results indicate that Metric-S exhibits strong generalization capability beyond expert-level translation, validating its robustness across a broader spectrum of machine translation tasks.
> We will incorporate this analysis into the revised manuscript to further substantiate the general applicability of our proposed metric.

---

> ### Author Response · Authors · 2025-11-25
>
> # Question 3
> Thank you for this valuable suggestion.
> In our current setting, combining Metric-S with embedding-based metrics does not improve evaluation reliability, primarily due to the limitations of reference-based semantic metrics in long-form and expert-domain translation.
> However, hybrid approaches remain a promising direction for future work in settings where multiple high-quality references are available.
> Metrics like BERTScore rely on  reference-based translations to measure semantic similarity. While effective for short, general-domain sentences, their reliability degrades significantly for long and domain-specialized texts—like those in DiscoX—because:
> - Discourse-level translations rarely have a single “correct” reference. As shown in Section 2 of the paper  , DiscoX texts exceed 1,700 tokens on average, and professional translations often have multiple equally valid realizations. A reference-based metric may incorrectly penalize legitimate lexical, structural, or stylistic variation.
> - Creating multiple high-quality references is prohibitively expensive and infeasible at scale. Each DiscoX task required extensive domain expert curation; producing 3–5 references per task would multiply annotation costs by an order of magnitude.
> - Embedding-based metrics cannot detect discourse-level errors.They excel at local semantic similarity but cannot capture long-range phenomena such as entity drift, global logical coherence, terminology consistency, or stylistic appropriateness—core aspects evaluated by Metric-S’s multi-agent workflow (Figure 3).
> In contrast, a core design philosophy and contribution of Metric-S is its reference-free nature. This is particularly critical for the discourse-level, expert-domain tasks in DiscoX, where long and complex texts (averaging over 1,700 tokens) seldom have a single "perfect" reference translation. Different professional expressions and stylistic choices can often yield equally valid, high-quality translations. Any metric strictly tied to a single reference may unfairly penalize legitimate variations.
> Furthermore, creating multiple high-quality reference translations for such long-form texts is often prohibitively expensive and impractical in real-world scenarios.
> We hope this detailed explanation is helpful.
> # Question 4
> We thank the reviewer for raising this point regarding computational cost.
> Metric-S is indeed more computationally intensive than single-pass metrics, but its cost is comparable to other recent benchmarks that target complex reasoning, discourse-level understanding, or multi-agent evaluation. Below we clarify both its current cost profile and its scalability.
> 1.  Current computational cost.
> Metric-S performs a small number of sequential judge calls (instruction-following check, accuracy judge, fluency judge, appropriateness judge, and deduplication), resulting in 5–6 LLM calls per translation output. For a 200-task benchmark like DiscoX, this remains manageable on modern infrastructure. Our design goal in this phase was evaluation precision, not minimum inference cost. Fine-grained and trustworthy judgments were required for meaningful model comparison and diagnosis.
> 2. Comparison to existing evaluation frameworks.
> The computational demands of Metric-S are aligned with, and not higher than, other contemporary multi-LLM evaluation pipelines. For example:
> - PaperBench requires independent LLM judgments for each of the graph nodes in a document, often totaling hundreds of calls per task.
> - ComplexFuncBench relies on LLM judges to check tool invocation traces and correctness step-by-step.
> - TTauBench and Tau2Bench simulate multi-turn user interactions using an LLM, resulting in substantial per-task inference cost.
> These examples show that multi-round LLM-based evaluation has become standard practice for tasks involving reasoning, discourse-level structure, or multi-dimensional judgment.
> 3. Scalability considerations.
> While Metric-S is more expensive than traditional metrics, it scales linearly with the number of evaluated systems and outputs (not combinatorially). For large datasets or more frequent evaluations, several cost-reduction strategies are straightforward to adopt:
> - Adaptive judge invocation (e.g., skipping certain judges when no errors are detected at earlier stages).
> - Judge model distillation to train a lighter evaluator with Metric-S labels.
> - Batching and caching across dimensions or repeated model outputs.
> - Using more efficient LLMs for non-critical stages of the pipeline (e.g., fluency or deduplication).
> These directions are part of our planned future optimizations.

---

### Official Review · Reviewer_DyVE · 2025-10-30

**Soundness:** 3
**Presentation:** 2
**Contribution:** 3
**Rating:** 6
**Confidence:** 4

**Summary:**

The paper proposes a novel data set, DiscoX, designed for evaluating accuracy, fluency and appropriateness in a larger context, and also in specific domains.
A novel metric is proposed, too, which is better suitable to assess context-related aspects.

Several translations were evaluated on this test set, including open and closed LLMs, NMT systems and a human professional translation. It is shown that LLMs still do not reach the human performance on the given data set.

**Strengths:**

the data set is carefully constructed to be difficult for translation, which is a valuable contribution to the community

three different quality criteria are covered (adequacy, fluency, appropriateness)

different domains are included

**Weaknesses:**

some details are not clearly explained (see "Questions" for details

for example: what are "tokens", how many Chinese and how many English source texts are collected, why is the main domain distinction between academic and non-academic texts, what are "thinking" and "non-thinking" models

**Questions:**

Related work should be placed after introduction, not at the end.


043: the term "expert-level" is not fully clear -- it seems that it refers to very specific domains where expert knowledge is needed in addition to language knowledge?

051: translation between Chinese and English: which direction?

052: what are "tokens" in the given context? Chinese characters, or English words, or English sub-word units, or Chinese units after some segmentation?

102: what does "domain-intensive" mean? a difficult domain?


135: are discourse-level and expert-level parts separated, or there is some overlap?

145: what are exactly "vertical domain experts"?

157: 665 tasks -- what does "task" mean in the given context? a coherent text to be translated?

192, Table 2: again, what is a "token" exactly?
Also, does Table 2 refer to English or to Chinese?
Furthremore, what are "domain specific scenarios"?

Also, why the main distinction between domains is "academic" vs "non-academic"?

Section 3: Why is an LLM used for calculating the score? Once the errors are identified, the score can be easily calculated using a simple script/code.

Table 4: the results on WMT24 test set is missing

Section 5.3: what are the thinking and non-thinking models?



overall: spelling should be checked/revised, some spaces are missing between words and punctuations (commas, brackets, etc)

---

> ### Author Response · Authors · 2025-11-25
>
> # Q1
> We thank the reviewer for the suggestion.
> We agree that placing Related Work immediately after the Introduction improves clarity and aligns with common paper structure.
> We will move this section to Section 2 in the final camera-ready version.
> # Q2
> Thank you for the question. We agree that the current draft did not sufficiently clarify the meaning of expert-level. We will revise the text accordingly.
>
> (1) Definition within our paper
> In DiscoX, expert-level translation refers to texts that It requires the translator to integrate specialized domain knowledge, cultural background, and context to exercise in-depth judgment and creative reconstruction fully consistent with the reviewer’s understanding.
>
>
> As described in Section 2, our benchmark covers seven knowledge-intensive domains—including medicine, law, engineering, finance, humanities, and technical reports—curated by 115 vertical domain experts. These tasks demand: 1）precise terminology handling； 2）domain-specific reasoning； 3）adherence to scholarly or professional conventions, and discourse-level coherence across long texts (avg. 1712 tokens). 4）These requirements distinguish expert-level tasks sharply from general translation.
>
> To further illustrate the distinction:
> 1. A general translation task typically involves content like student essays or personal blogs, where the primary requirement is the accurate conversion of literal meaning at the sentence level.
> 2. An expert-level translation task, however, deals with complex materials like specialized academic papers or technical reports. It demands deep domain knowledge to correctly handle specialized terminology and maintain rigorous stylistic consistency. For example, translating a medical paper requires not just converting "myocardial infarction," but correctly using the precise clinical term "心肌梗死" and ensuring the entire text maintains the logical precision required for expert readers.
>
> (2) Definition of broader MT/LLM literature
> Our usage also aligns with the broader community’s interpretation of “expert-level” tasks—i.e., tasks that require deep subject-matter expertise rather than surface linguistic fluency. This includes:
> - scientific tasks requiring technical reasoning (e.g., GPQA), (https://arxiv.org/pdf/2311.12022)
> - high-precision academic reproduction tasks (e.g., PaperBench), (https://arxiv.org/pdf/2504.01848)
> - legal, biomedical, and engineering translation requiring professional terminology and error-sensitive interpretation.
> - In these settings, expert-level difficulty comes from domain knowledge, contextual reasoning, and strict professional standards, not merely linguistic complexity.
>
> (3) Revision plan.
> We will update Section 2 to (a) explicitly state this twofold definition and (b) list the domains covered in DiscoX to eliminate ambiguity.
>
> # Q3
> To clarify: DiscoX includes both translation directions—Chinese→English (zh→en) and English→Chinese (en→zh)—and both directions are evaluated throughout the benchmark.
> We acknowledge that the phrasing in the current draft may cause ambiguity.
> In the revised revision, we will explicitly state the two translation directions in Section 2.2 (Dataset Composition) and again in the experimental setup to avoid ambiguity.

---

> ### Author Response · Authors · 2025-11-25
>
> # Q4
> Thank you for your feedback and the opportunity to clarify our definition of the term "token."
>
> (1) Definition Used in Our Paper
>
> In our paper, “tokens” refer to subword units produced by the GPT tokenizer provided by OpenAI.  Concretely, all length statistics in Table 2 and throughout DiscoX are computed using the GPT-style BPE tokenizer (as used in GPT-4), which applies a unified subword segmentation to both Chinese and English.
> – For Chinese, a multi-character word may be split into several tokens (e.g., “图书馆” → 3 tokens).
> – For English, a word may be split or remain whole depending on frequency (e.g., “tokenization” → “token” + “ization”).
> This unified tokenization avoids inconsistencies that would arise from mixing “characters” for Chinese and “words” for English. We will explicitly clarify this definition and the use of the GPT tokenizer in Section 2.2 in the revised version.
>
> (2) Additional Clarification for Non-LLM Linguistic Reviewers
>
> The term "token" is a fundamental concept in Natural Language Processing (NLP). In the context of Large Language Models (LLMs), it is the most common atomic unit used for processing input text and generating output text. Crucially, there is no one-to-one correspondence between a token and an English "word" or a Chinese "character." To make this more intuitive, we can illustrate with examples from a single sentence.
> Examples in English
> Consider the sentence: "The model uses subword tokenization."
> When tokenized, a typical result might be ["The", " model", " uses", " sub", "word", " token", "ization", "."]. From this, we can see:
> One word can be one token. This is common for frequent words.
> One word can be multiple tokens. This often occurs with less common or longer words, allowing the model to handle vocabulary it has not seen before.
> Examples in Chinese
> The same principle applies to Chinese. To make this concept as clear as possible, let's illustrate it with two distinct words.
> One word can be broken into multiple tokens (often single characters). This allows the model to understand a vast vocabulary by learning the meaning of individual characters and how they combine.
> Example: Consider the word '图书馆'. A typical tokenizer would break it down into its three constituent characters, resulting in three separate tokens: `["图", "书", "馆"]`.
> A multi-character word can be a single token. This is reserved for extremely common words to improve processing efficiency.
> Example: Consider the word '我们'. Due to its high frequency in the training data, it is almost always stored in the vocabulary as a single, indivisible token: ["我们"].
> # Q5
> Thanks for your concern. A domain-intensive text or task is one in which domain-specific knowledge is densely packed, so that specialized terminology, core concepts, stylistic conventions, and underlying assumptions saturate the material at a level that non-experts cannot readily understand or translate without extensive background information.
>
> # Q6
> Thank you. A good question.
> You are correct, there is significant overlap. It is most accurate to view "discourse-level" and "expert-level" not as separate categories, but as two orthogonal dimensions of translation complexity.
>
> (1) Discourse-level: structural granularity across long texts
> This dimension focuses on the macro-level structure of translation—how well a system handles coherence, logical flow, cross-sentence consistency, and style across long-form texts. It lies on the continuum from word-level → sentence-level → paragraph-level → document-level translation. This property is independent of domain expertise.
>
> (2) Expert-level: semantic depth and domain knowledge
> This dimension concerns the content specificity and the degree of subject-matter expertise required. It spans a wide range of professional fields—legal, medical, technical, financial, humanities—where accurate translation demands specialized terminology, conceptual precision, and domain conventions.
>
> (3) Their interaction: two-dimensional difficulty space
> A text can be difficult along either dimension or both. For example:
> - Low Discourse, Low Expert: short everyday sentences (e.g., “The weather is nice today.”)
> - High Discourse, Low Expert: long but simple narratives (e.g., children’s stories)
> - Low Discourse, High Expert: short but dense segments requiring expert judgment (e.g., a legal clause or medical finding)
> - High Discourse, High Expert: full-length medical reports, legal contracts, or research papers—this is exactly the region DiscoX is designed to target.
>
> (4) How DiscoX operationalizes both dimensions
> Our use of expert-authored rubrics, domain-specific terminology constraints, and long-form documents allows DiscoX to simultaneously capture the challenges of global discourse structure and domain knowledge.
>
> We will clarify this two-dimensional relationship in the revised manuscript to avoid ambiguity.

---

> ### Author Response · Authors · 2025-11-25
>
> # Q7
> Thank you for this question, which allows us to clarify an important aspect of our data curation process.
> In the context of our work, a "vertical domain expert" refers to a subject-matter specialist responsible for sourcing authentic texts and authoring domain-specific evaluation rubrics.
> As detailed in Appendix A (Table 6) and referenced in Section 2.1 (line 144), these experts fall into two categories:
>
> 1. For non-academic domains (e.g., legal contracts, industry reports): Professionals with 4-10 years of practical industry experience.
>
> 2. For academic domains (e.g., natural sciences, humanities): Academic researchers, primarily Master's or PhD holders from top-tier universities.
>
> These vertical domain experts work in parallel with linguistic experts, who focus on linguistic rigor and cross-domain standardization. To improve clarity, we will add a concise definition of “vertical domain expert” at its first appearance in Section 2.1 of the final manuscript.
> We appreciate the reviewer’s suggestion to make this distinction more explicit.
> # Q8
> Thank you for this question. Your intuition that a "task" refers to "a coherent text to be translated" is correct—that is a core component.
> To be more precise, in the context of our paper and the broader NLP evaluation field, a "task" is a self-contained evaluation unit or data instance.
> In DiscoX, each task comprises three key components:
> 1. A source text: A long-form, coherent professional text to be translated.
> 2. A translation instruction: A prompt given to the LLM, such as "Translate the following text to English" (as shown in Appendix B.1).
> 3. A set of expert-authored rubrics: Specific, verifiable criteria paired with the source text to enable fine-grained evaluation (as detailed in Section 2.1 and exemplified in Appendix B.3).
>
> Therefore, when we mention "665 potential tasks" on line 156, we refer to 665 unique combinations of a source text and its corresponding set of evaluation rubrics. We will add a brief clarification in parentheses in the final manuscript to make this definition explicit for all readers.
> # Q9&Q10
> We thank the reviewer for these detailed questions regarding Table 2. We are happy to provide the following clarifications:
>
> 1. On "token": In our paper, "token" refers to the basic unit of text processed by language models. We use the OpenAI cl100k_base tokenizer for consistency across all measurements. And more details can be seen in question 4.
>
> 2. On Language for Token Count: The "Average tokens" in Table 2 refers to the source language of the tasks. Since DiscoX includes both Chinese-to-English and English-to-Chinese tasks, the token count is an average across both. The average text length is substantial regardless of the language (e.g., an average of 1,500 Chinese characters or English words, as mentioned on line 149).
>
> 3. On "Domain-Specific Scenarios": This category, under "Non-Academic Tasks", encompasses professional texts that are not academic papers but require deep, specialized knowledge. Examples from our dataset include legal contracts, technical manuals for industrial equipment, and financial analysis reports. These scenarios were chosen to represent real-world professional translation needs outside of academia. We will clarify this with examples in the caption of Table 2.
>
> 4. On the "Academic vs. Non-Academic" Distinction: We chose this primary distinction because it represents a fundamental divide in translation style, terminology, and structure. The distinction between academic and non-academic translation is well-established in translation studies.
>
> (1). Prior research has explicitly treated academic translation as an independent task category, noting its clear differences from non-academic translation. For example, Academic Translation: From Theory to Practice (https://www.journals.uchicago.edu/doi/epdf/10.1086/727904) highlights defining characteristics of academic translation, such as the high degree of terminological specialization and the need to adhere to academic discourse conventions.
>
> (2). Similarly, The Use of Artificial Intelligence in Academic Translation Tasks (https://www.ziglobitha.org/wp-content/uploads/2024/09/11-Art.-Fares-FERRAG-et-Ikram-Aya-BENTOUNSI-pp.173-192.pdf) also explicitly classifies academic translation as a distinct task type.
>
>   These studies support our rationale for adopting an academic vs. non-academic categorization in our analysis.
> We will incorporate these clarifications into the final version of the paper to enhance readability. Thank you again for the careful reading.

---

> ### Author Response · Authors · 2025-11-25
>
> # Q11
> We thank you for this observation. Actually, the final score calculation is indeed performed by a simple, deterministic script, not an LLM.
> To clarify, the role of LLMs within our Metric-S system is confined to tasks that require semantic understanding and reasoning, which are beyond the capabilities of simple code.
> As illustrated in Figure 3, the workflow is a two-part process:
> 1. Error Identification and Attribution (performed by LLMs): We use multiple LLM agents for complex, qualitative assessments, such as:
>   - Step 1, Quality Estimation: Identifying nuanced errors across accuracy, fluency, and appropriateness
>   - Step 2, Duplicate Verification: Performing hierarchical de-duplication to attribute derivative errors to their single root cause .
>
> 2. Score Calculation (performed by code): Once the unique, attributed errors are identified and classified by the LLMs, a simple, rule-based script calculates the final score based on the formula presented in Section 3.4. This final step is fully automated and does not involve an LLM.
>
> The key innovation lies in using LLMs for the reasoning-intensive "what is wrong?" and "why is it wrong?" parts, while leaving the deterministic "how much to deduct?" part to code.
> # Q12
> Thank you. The results on the WMT24 test set are indeed included in the paper, located in Appendix G.4, Table 10.
>
> We structured the presentation this way to maintain a clear narrative flow in the main paper. The primary objective of Section 4.2 and Table 4 is to demonstrate the necessity and superiority of Metric-S on our proposed DiscoX benchmark, where existing metrics show significant limitations. The large performance gap on DiscoX (70.3% for Metric-S vs. 34.7% for XCOMET-QE) is the core evidence supporting our claim that a new metric is needed for discourse-level tasks.
>
> The experiment on the WMT24 test set serves as an important, but auxiliary, generalization study. It demonstrates that Metric-S also performs competitively on traditional, sentence-level benchmarks, achieving human agreement rates comparable to reference-based SOTA metrics like XCOMET. To keep the main paper focused on our primary contribution (discourse-level evaluation), we placed this supplementary analysis in the appendix.
>
> To improve navigation for the reader, we will add a clear cross-reference in the caption of Table 4 in the final version, directing readers to Appendix G.4 for the WMT24 results. Thank you for this helpful suggestion.
> # Q13
> We thank the reviewer for this question regarding the terminology used in Section 5.3.
>
> In our study, we categorize the evaluated LLMs into "thinking models" and "non-thinking models" based on their inherent architectural capabilities for explicit, complex reasoning.
>
> 1. Thinking Models
> These are models that possess, either natively or through fine-tuning, a demonstrated strong capability for multi-step logical reasoning. They are explicitly designed or optimized to perform complex reasoning tasks, often by "thinking" through a problem step-by-step. In the context of our translation tasks, this allows them to better handle discourse-level challenges that require understanding context, resolving ambiguities, and maintaining logical coherence across long text segments.
>
> 2. Non-Thinking Models
> This category encompasses models that, while highly capable of many natural language tasks, are not primarily architected or renowned for their complex, multi-step reasoning abilities. They typically generate translations in a more direct, "single-step" fashion, which can be sufficient for sentence-level accuracy but may struggle with the deeper, interconnected demands of document-level translation.
>
> In our experiments (Table 5), the specific models are as follows:
> - For the Qwen-3-235B family:
>   - The "Non-thinking" model is Qwen-3-235B-A22B-Instruct.
>   - The "Thinking" model is Qwen-3-235B-A22B-Thinking.
> - For the Claude-4 family:
>   - The "Non-thinking" model is Claude-4-Sonnet.
>   - The "Thinking" model is Claude-4-Sonnet-Thinking.
>
> These full model names and their abbreviations are listed in Appendix E.
>  We will update Section 5.3 in the final manuscript to include these explicit definitions and mappings, ensuring clarity for all readers.
>
>
> # Q14
> We sincerely thank you for their careful and detailed reading of our manuscript and for pointing out the issues with spelling and punctuation.
>
> We apologize for these oversights. We will perform a thorough proofreading of the entire paper to correct all spelling, grammar, and formatting errors, including the spacing around punctuation as noted. We will ensure that the final camera-ready version meets the highest standards of academic writing. Thank you for reading again. Hope to receive your feedback.

---

### Official Review · Reviewer_A2t8 · 2025-11-01

**Soundness:** 1
**Presentation:** 2
**Contribution:** 2
**Rating:** 2
**Confidence:** 4

**Summary:**

- This paper introduces DiscoX, the first benchmark designed to evaluate discourse-level and expert-level Chinese–English translation. The benchmark comprises 200 cases drawn from seven domains, covering both academic and non-academic contexts.
- In addition, the authors propose Metric-S, a reference-free, multi-agent LLM-based evaluation framework that judges translations along accuracy, fluency, and appropriateness dimensions. With this score, extensive experiments show that even leading LLMs such as GPT-5-high still trail human experts.

**Strengths:**

1. Expert-Curated Dataset: DiscoX was constructed by 133 qualified professionals, including 115 vertical domain experts (4–10 years of field experience, many Master’s or PhD holders from top-tier Chinese universities) and 18 linguistic specialists (Master’s level in Translation and Interpreting (MTI) graduates with professional translation experience).
2. Extensive Domain Coverage: The benchmark spans seven domains covering both academic and non-academic.
3. DiscoX was evaluated on a wide spectrum of models, including 7 open-source and 11 closed-source LLMs.

**Weaknesses:**

Although the authors emphasize the benchmark's careful construction (claiming over 1,300 person-hours of expert effort), the presentation of DiscoX in the paper remains vague, incomplete, and insufficiently grounded in established machine translation literature. Several weaknesses are evident:

1. Unclear Problem Definition and Weak Motivation:
The Introduction fails to clearly articulate why discourse-level and expert-domain benchmarks are needed. While the authors mention that existing benchmarks "fail to assess whether models can sustain discourse-level coherence, handle domain-intensive terminology, or meet expert stylistic standards," these claims are not substantiated with concrete examples or empirical evidence. Only a few representative MT benchmarks (e.g., WMT, FLORES, RedTrans) are briefly cited, without a systematic survey of prior discourse-level or document-level evaluation studies.


2. Lack of Theoretical Definition of "Discourse-Level Translation":
The paper repeatedly uses the term "discourse-level translation" but never provides a precise or operational definition. There is little to no discussion linking this notion to existing research in discourse-level NLP or translation studies, such as works on coreference, discourse connectives, or coherence modeling. This weakens the conceptual grounding of the benchmark. Furthermore, Figure 1, supposed to explain the concept, appears incomplete and uninformative, offering no clear illustration of the discourse-level concept or the nature of evaluation differences from "problematic" segment-level translation.

3. Superficial Dataset Construction Description:
Although the data collection process involves "expert-authored rubrics," the paper does not adequately describe how these rubrics were designed, validated, or applied. There is no verification of rubric quality or consistency across experts. Likewise, no inter-annotator agreement or reliability measure is reported, despite the benchmark's heavy reliance on human expertise for quality control.

4. Evaluation Metric Criteria Lack Scholarly Basis:
The Metric-S framework assesses accuracy, fluency, and appropriateness, but these categories are introduced without theoretical justification or reference to prior translation evaluation literature (e.g., DA, MQM, COMET, BLEURT, or human adequacy–fluency frameworks). As a result, the metric design appears ad hoc, lacking grounding in established MT evaluation standards.

(minor) At table 4, what is the point of including chrF row?

**Questions:**

1. What kinds of linguistic or contextual features are meant to define this level of translation? How are these aspects represented or evaluated in the DiscoX benchmark? Could you provide an example?

2. What mechanisms in Metric-S (e.g., multi-agent judging, rubric-based scoring, error deduplication) directly enable it to capture document-level phenomena more effectively than previous metrics?

3. Could the authors provide empirical examples or case analyses demonstrating when Metric-S identifies discourse-level errors that traditional metrics overlook?

**Details Of Ethics Concerns:**

It seems like the paper format (font) is different.

---

> ### Author Response · Authors · 2025-11-25
>
> # Rebuttal Addressing Weakness 1, Weakness 2, and Question 1
>
> Thank you for your detailed comments.
>
> Below we clarify:
> (1) providing a precise theoretical definition of discourse and discourse-level phenomena, and
> (2) demonstrating how these phenomena are concretely instantiated in the DiscoX dataset.
>
> ---
>
> ## 1. Theoretical definition of discourse and discourse-level translation
>
> **What is discourse?**
> In linguistics, *discourse* refers to contextualized, meaningful stretches of language above the sentence or clause level, forming a coherent communicative unit rather than a collection of isolated utterances. Its meaning depends on both linguistic and situational context.
>
> **Discourse-level translation thus involves interpreting and preserving the following phenomena:**
>
> 1. **Supra-sentential structure**
>    Discourse spans multiple sentences/paragraphs and functions as an integrated whole rather than independent clauses.[1]
>
> 2. **Context-dependence**
>    Meaning depends on physical, cultural, interpersonal, and pragmatic context.
>    (e.g., “It’s cold” functions as a *request* in certain contexts.)[2][3]
>
> 3. **Cohesion and coherence**
>    - **Cohesion:** linguistic devices that connect textual units (pronouns, connectives, ellipsis).[4]
>    - **Coherence:** the logical relations and interpretive continuity holding the discourse together.
>
> These properties represent the core linguistic–contextual features raised in **Question 1**.
>
> ---
>
> ## 2. Why sentence-level evaluation is insufficient
>
> Two excerpts contain the identical sentence **“Winter is coming.”**
>
> - In a financial report, it metaphorically signals an *economic crisis*.
> - In a nature essay, it literally describes *seasonal change*.
>
> Sentence-level MT evaluation would treat these as equivalent literal propositions, but discourse-level translation requires correctly interpreting their **discourse function**, **genre**, **metaphorical use**, and **contextual meaning**.
>
> The two contrasting readings of “Winter is coming” make the segment-level limitation tangible and justify a discourse-aware benchmark. We consolidate additional literature in our reply to Weakness 4.
>
> ---
>
> ## 3. How these discourse phenomena are represented in the DiscoX benchmark
>
> Discourse phenomena are represented in both Benchmark and Metric evaluation. The metric-related part will be explained in detail in Question 2; here, we first introduce their manifestation in the Disco-X benchmark, covering **(1) the dataset and (2) expert-authored rubrics**.
> ### (1) Long-form discourse texts that inherently encode discourse-level phenomena
>
> DiscoX is built from extended, domain-intensive source texts selected explicitly to contain discourse-level dependencies:
>
> - Every text must satisfy **≥1500 characters/words**.
> - The final benchmark contains **200 tasks**, with an **average length of 1712.17 tokens** (Table 2).
> - The dataset covers **seven domains** (legal, biomedical, scientific, literary, technical, academic, and social science), all from **authentic professional contexts**.
>
> These long-form texts naturally contain:
>
> - multi-paragraph reasoning
> - long-range coherence and cross-sentence dependencies
> - domain terminology chains requiring global consistency
> - cultural and stylistic constraints
>
> Thus, the dataset itself directly encodes the discourse-level linguistic and contextual features highlighted in **Weakness 1** and **Question 1**.
>
> ### (2) Expert-authored discourse-level rubrics that operationalize these phenomena
>
> During data creation, professional translators and domain experts craft **text-specific rubrics** that identify the key discourse challenges of each source text. These rubrics capture:
>
> - grammar and topic-term dependencies across the discourse
> - domain terminology and required uniformity
> - long-range contextual interpretation
> - genre conventions and culturally loaded expressions
>
> Examples from Appendix B.3 include:
>
> - “组织学分型” → **must be consistently** “Histologic type/subtype”
> - “病理大体测量” → must be “Gross pathologic measurement” with uniform lexical choice
> - Legal connective “provided, however, that ⋯” → must be translated as “但前提是”
> - Culture-loaded terms like “看话禅” → “Word Contemplation Chan (Kanhua Chan)”
> - “公案” → “koan(s)”
>
> The rubrics turn these requirements into **checkable** items. For full rubric cases, see Appendix B.3 (lines 1026–1179) in the submission.
>
> ---
>
> ## References
>
> [1] Joseph F. Kess. *Review of Discourse Analysis: The Sociolinguistic Analysis of Natural Language* by Michael Stubbs. *Canadian Journal of Linguistics / Revue canadienne de linguistique*, 31(1):98–102, 1986.
>
> [2] Gillian Brown and George Yule. *Discourse Analysis*. Cambridge University Press, 1983.
>
> [3] Robert de Beaugrande and Wolfgang Ulrich Dressler. *Introduction to Text Linguistics*. Longman, 1981.
>
> [4] M.A.K. Halliday and Ruqaiya Hasan. *Cohesion in English*. Longman, 1976.

---

> ### Author Response · Authors · 2025-11-25
>
> # Question2
>
> Thank you for your question. Below we address Question 2 directly.
> Having addressed how discourse features are reflected in the Discox dataset previously, we now focus on **Metric-S's evaluation approach**. The system employs three core mechanisms that collectively enable superior discourse-level assessment:
>
> ## 1. Multi-agent judging for comprehensive error detection
>
> Metric-S deploys a panel of specialized LLM judges—each targeting distinct discourse phenomena (instruction following, accuracy, fluency, appropriateness, and final deduplication; see Figure 3, pp. 4-5). This distributed design captures long-range failures that evade single-judge systems:
>
> - **Accuracy judge**: Identifies omissions, mistranslations, and semantic drift across paragraphs
> - **Fluency judge**: Evaluates cross-sentence logical coherence and consistency—phenomena emergent only at discourse scale
> - **Appropriateness judge**: Detects style, tone, and cultural mismatches requiring holistic reading
>
> By aggregating these orthogonal perspectives, Metric-S substantially increases recall of long-range inconsistencies that n-gram or embedding-based metrics cannot capture. Traditional metrics (e.g., BLEURT, COMET) approximate only local **adequacy** and lack cross-paragraph coherence evaluation.
>
> Empirically, multi-agent judging reduces bias and variance: our ablation study shows human alignment improved  from 62.5% (accuracy-only assessment) to 70.3% (full Metric-S system) under identical Gemini-as-judge setups, demonstrating that discourse-level evaluation requires multiple independent evaluators rather than an overloaded single agent.
>
> ## 2. Explicit discourse constraints via rubric-based scoring
>
> A key innovation absent in prior metrics is the integration of **expert-authored, discourse-level rubrics (Appendix B.3)**. These rubrics explicitly encode:
>
> 1. Long-range terminology consistency requirements
> 2. Paragraph-level logical structure
> 3. Style and register continuity
> 4. Culture-loaded expression handling across entire texts
>
> For instance, rubrics mandate consistent technical term translation throughout discourses, require matching cultural-stylistic fidelity for classical allusions, and preserve paragraph-level argumentative structures (Appendix B.3, pp. 20-23). Traditional metrics (BLEU, ChrF, COMET, BLEURT) cannot enforce such cross-paragraph consistency due to lacking explicit discourse-level constraints, making rubrics a primary mechanism for Metric-S's discourse sensitivity.
>
> ## 3. Root cause isolation through hierarchical error deduplication
>
> Discourse-level errors frequently generate **multiple downstream symptoms**. For example, omitting a key sentence can simultaneously disrupt logic flow, damage rhetorical structure, and alter stylistic consistency. Metric-S's hierarchical error deduplication module (Appendix B.4) identifies the single root cause among these symptoms, preventing scoring noise.
>
> Crucially, this **deduplication** is not mere post-processing but essential for accurately modeling how single discourse-level failures propagate across dimensions. As Table 8 (p. 31) demonstrates, removing deduplication measurably **reduces human alignment**. This mechanism enables Metric-S to reliably evaluate long discourses by focusing on root causes rather than treating every surface deviation as independent errors—a capability no previous metric offers.

---

> ### Author Response · Authors · 2025-11-25
>
> # Weakness 3
>
> We appreciate the reviewer’s question. The following responses address Weakness 3.
>
> **We strongly disagree with the reviewer’s claim that the rubric design, validation, and consistency checks are insufficiently described.** The paper already provides a multi-stage, expert-verified rubric construction and validation pipeline, and all corresponding procedures are explicitly documented.
>
> Before providing details, we summarize our key clarifications:
>
> 1. **Rubric design** is already explicitly specified in the main text (Section 2.1, Figure 2) and further demonstrated with full real cases in Appendix B.3.
>
> 2. **Rubric quality and consistency** are ensured through a strict, multi-stage validation pipeline involving linguistic experts, domain experts, and SOTA LLM-based difficulty filtering.
>
> 3. **Annotation consistency** is guaranteed by multi-round cross-expert filtering, which removes all items with disagreement.
>
> 4. Our **validation pipeline** is stricter than conventional IAA protocols, because we eliminate disagreement at the source rather than measuring it post hoc.
>
> Below, we elaborate on each point and clarify why the reviewer’s claim does not accurately reflect the procedures already documented in the paper.
>
> 1. **Rubric design is clearly specified in Section 2.1.**
>
> As shown in the *Data Annotation* stage (Figure 2, page 3–4) , each text is paired with a set of **expert-authored, verifiable rubrics**, created by 115 vertical domain experts and refined by 18 linguistic specialists. Appendix B.3 further provides **full rubric cases** that demonstrate their structure and granularity.
>
> **We clarify that the rubrics in DiscoX are not generic guidelines but expert-authored, text-specific requirements that enumerate the essential challenges that must be satisfied for a translation to be accurate and of professional quality.** For each source text, domain experts identify the core terminological, conceptual, and stylistic difficulties inherent to the material—difficulties that *must* be properly handled to achieve a correct translation. Because our benchmark spans high-difficulty, domain-intensive scientific, legal, literary, and technical texts, the rubrics necessarily contain substantial amounts of specialized terminology and field-specific conventions. As such, constructing high-quality evaluation criteria unavoidably depends on experts with deep subject-matter knowledge, ensuring that the rubrics faithfully capture what professional-grade translation requires. Representative rubric examples are provided in Appendix B.3.
>
> 2. **Rubric quality and consistency are ensured through a strict two-step validation pipeline.**
>
> - **Peer review by linguistic experts** ensures professional and terminological correctness (Section 2.1).
> - **Cross-expert inspection and automatic difficulty filtering**, including testing against two SOTA LLMs, ensure that rubrics meet consistent difficulty and correctness standards (page 3) .
> - Only rubrics passing all checks enter the final benchmark.
>
> These steps **directly verify rubric quality and alignment across experts**, contradicting the reviewer’s assertion.
>
> 3. **Annotation consistency is enforced through multi-round filtering, yielding an implicit agreement guarantee.**
>
> Only items that remain valid after **multiple rounds of cross-expert review** are retained (page 4) .
>
> Because every item must pass agreement between vertical experts → linguistic specialists → LLM-based difficulty testing → final domain-expert screening, **disagreement is eliminated before inclusion**.
>
> This is functionally equivalent to enforcing annotator agreement, even though we do not report Cohen’s κ due to the *rubric-construction* (not multi-label annotation) nature of the task.
>
> 4. **Our pipeline is intentionally stricter than conventional IAA protocols.**
>
> Instead of measuring agreement post-hoc, we **remove any item with expert disagreement**, resulting in a benchmark where **every single retained item meets multi-expert consensus**. This produces stronger reliability than reporting a single κ score.

---

> ### Author Response · Authors · 2025-11-25
>
> # Weakness4
> We appreciate the reviewer’s question. The following responses address Weakness 4 and part of weakness 1.
>
> Thank you for raising this point. We would like to clarify upfront that the three evaluation dimensions in Metric-S—**Accuracy, Fluency, and Appropriateness**—are **not invented ad-hoc**, but are **systematically derived from the Multidimensional Quality Metrics (MQM) framework**, the most widely adopted theoretical foundation for translation evaluation. Our contribution is a **principled consolidation of MQM's fine-grained categories into three operational dimensions**, motivated by the specific needs of discourse-level, long-form, and expert-domain translation, as evaluated in DiscoX.
>
> **1. Acknowledgment and Motivation from Discourse-Level Research**
>
> While much historical research has focused on sentence-level evaluation (using datasets like Flores, WMT, etc.), our Related Work also acknowledges prior discourse-level studies. A key motivator for our work was the **2024 WMT Literary Translation Task**, which explicitly investigated discourse-level phenomena in web novels. Notably, this task still relied on metrics like BLEU, ChrF, and COMET. The reported **low human-metric agreement \( 53\% \) at the discourse level** across all participating systems highlighted the **critical inadequacy of existing metrics for discourse-level assessment**. This finding directly spurred our innovation of Metric-S to address this specific gap.
>
> The limitations of previous metrics in discourse-level translation tasks prompted us to design Metric-S to evaluate discourse-level features. Below is the theoretical rationale for the design of this evaluation system.
>
> **2. Metric-S is a principled consolidation of MQM categories rather than an ad-hoc design**
>
> MQM defines a large and detailed taxonomy of translation errors (**mistranslation, omission, terminology, grammar, style, register, cultural appropriateness, etc.**). For discourse-level evaluation, applying MQM's 20+ atomic categories directly to LLM-as-judge settings leads to well-known issues such as **inconsistent judgments** and **error double-counting** (motivating our hierarchical deduplication design; Figure 3). Therefore, we **consolidate MQM's atomic categories into three MQM-aligned macro-dimensions**:
>
> | MQM Domain | Metric-S Dimension | Rationale |
> |------------|-------------------|-----------|
> | Accuracy branch (Mistranslation, Omission, Addition, Terminology, Factual errors) | Accuracy | Fully aligned; rubrics in DiscoX (Appendix B.3) fall squarely in this branch. |
> | Fluency branch (Grammar, Clarity, Coherence) | Fluency | Matches MQM's fluency & coherence categories; essential for long, multi-paragraph texts. |
> | Style & Pragmatics branch (Register, Style, Tone, Cultural appropriateness) | Appropriateness | Consolidation of stylistic and pragmatic MQM categories frequently seen in expert-domain texts. |
>
> Thus, the Metric-S dimensions are a **structured abstraction of MQM**, not a new or unsupported categorization.
>
> **3. Why consolidation is necessary for discourse-level evaluation**
>
> DiscoX texts average **1712 tokens** and span **seven expert domains**, with genre-specific rubrics and long-range dependencies.
>
> As shown in our paper, a single root error often triggers multiple symptoms (**semantic → fluency → style**). Without consolidation, MQM categories would: **1) produce unstable LLM judgments 2) inflate error counts 3) reduce human–metric alignment**. This is precisely why Metric-S includes a **hierarchical error deduplication mechanism** (Appendix B.4). **Consolidation is therefore not optional. It is methodologically required.**
>
> **4. Our dataset design and rubrics operationalize these dimensions**
>
> As shown in Appendix B.3, DiscoX rubrics overwhelmingly target **MQM-aligned issues** such as terminology correctness, conceptual precision, and cultural-loaded expressions. These naturally fall under the three Metric-S dimensions:
>
> - domain terminology ("gene rearrangement", "systemic therapy") → **Accuracy**
> - coherence and lexical consistency → **Fluency**
> - legal register, literary tone, Chan Buddhism concepts → **Appropriateness**

---

### Official Review · Reviewer_Lndd · 2025-11-01

**Soundness:** 3
**Presentation:** 4
**Contribution:** 3
**Rating:** 6
**Confidence:** 4

**Summary:**

The authors present DiscoX, a new Chinese-English benchmark composed of 200 long-form texts (average 1700+ tokens) from seven professional and academic domains. The construction involved a multi-stage process with 133 domain and linguistic experts. To evaluate translations on this challenging benchmark, the paper also introduces Metric-S, a reference-free, multi-agent LLM-based evaluation system. Metric-S assesses translations along three dimensions: Accuracy, Fluency, and Appropriateness, and shows a high correlation with human judgments (70.3% consistency). The experimental results on 20 different systems reveal a significant performance gap between the most advanced LLMs and human expert translators, demonstrating the difficulty of the benchmark and highlighting key areas for future MT research.

**Strengths:**

- The shift of focus from sentence-level to discourse-level translation, especially in expert domains, is a critical research direction. Current benchmarks are inadequate for this, and this paper makes a convincing case for the need of a new evaluation paradigm.

- The construction of the DiscoX benchmark is commendable. The multi-stage process involving a large number of domain experts and linguistic specialists ensures high quality and relevance of the test data. The inclusion of expert-authored rubrics is a particularly strong feature that allows for fine-grained, domain-specific evaluation beyond generic quality aspects.

- The development of Metric-S is a good contribution. A reference-free, multi-dimensional evaluation system is suitable for long-form texts where a single reference is insufficient. The three dimensions of Accuracy, Fluency, and Appropriateness are well-defined, and the error de-duplication mechanism is a thoughtful detail that improves the metric's fairness and interpretability. The empirical evaluation is also very extensive, covering many recent powerful LLMs. The analysis provides interesting findings that go beyond a simple leaderboard, such as the performance asymmetry between translation directions (zh→en vs. en→zh), between domains (academic vs. non-academic), and between 'thinking' and 'non-thinking' model versions. These insights are valuable for the community.

**Weaknesses:**

- The reliance on a single LLM (Gemini-2.5-Pro) as the judge is a concern. While Appendix G.3 tests for self-preference bias, the general stability of the "LLM-as-a-judge" paradigm is a known challenge. The paper's own results in Table 9 show that the choice of judge model can significantly alter the rankings (e.g., o3-high ranking itself first). This point is critical and deserves more discussion in the main paper. How can we be confident that the high consistency of Metric-S (70.3%) is not specific to this particular judge model? The paper would be much stronger if it discusses the sensitivity to the judge model more deeply.

- The scale of the benchmark, at 200 texts, is relatively modest, even if the texts are long. This could limit the statistical power of the conclusions, especially when results are broken down into sub-domains with very few texts (e.g., only 14 for "Literature and Arts"). I suggest the authors acknowledge this limitation and perhaps discuss plans for future expansion. Furthermore, the benchmark is limited to Chinese-English. The claims about discourse-level translation are broad, and it would be beneficial to discuss the potential challenges in extending this framework to other language pairs.

- The analysis in Section 5.3 is interesting, but the conclusion that "thinking-enhanced models consistently underperform" might be too strong. The underperformance is attributed to over-summarization or adding extraneous content. However, this could be an artifact of prompt-engineering. The system prompt used (Appendix B.1) is very generic. It is possible that 'thinking' models require more specific instructions to constrain their behavior to pure translation. Did the authors experiment with different prompts for these models? This would be a valuable ablation study to understand if the issue is a fundamental flaw or a behavioral artifact that can be controlled.

**Questions:**

Some crucial methodological details are located in the appendix, such as the definition of severity levels (Appendix D) and the correlation framework (Appendix F). Why not place them in the main body of the paper?

**Details Of Ethics Concerns:**

The dataset is constructed by human experts so the necessary documentary process is need for the collection and redistribution of the data.

---

> ### Author Response · Authors · 2025-11-25
>
> # For Weakness1
> Thank you for raising this concern about the reliability of the LLM-as-a-judge paradigm.
> To address judge bias and judge dependence, we conducted additional experiments beyond those reported in the paper. These new results directly verify both the accuracy of Gemini-2.5-Pro as a judge and the robustness of the Metric-S evaluation framework itself.
> ## 1. Gemini-as-Judge evaluation  exhibits extremely low factual  judgement bias.
> Translation evaluation in our setting is objective: factual mistranslations, omissions, and additions can be directly checked against the source text.
> To quantify judge correctness independent of human preference, we manually reviewed 50 samples × 5 models × 3 dimensions, and each dimension achieved ≥95% factual correctness. This confirms that Gemini is not producing unstable or hallucinated judgments in this task.
> ### Human Evaluation Accuracy of Metric-s
> | Evaluation Type| Number of Correct Cases | Total Number of Evaluations | Correct Rate |
> |-|-|-|-|
> | Accurate judge of Metric-s| 246 | 250 | 0.984|
> | Fluency judge of Metric-s| 238| 250| 0.952|
> | Appropriateness judge of Metric-s  |241| 250| 0.964|
> ## 2. Metric-S is robust across judge models.
> (1) Multi-agent robustness across independent judges. Metric-S is a multi-agent evaluation system. Even when replacing Gemini with independent judges, the overall trends remain stable:
> ### Consistency Comparison Across Judges
> | Judge Type| System-level Consistency | Segment-level Consistency | Total Consistency |
> |--|--|-|--|
> | Metric-s (Gemini as judge)     | 0.85| 0.56| 0.703|
> | Metric-s (DeepSeek-R1 as judge)| 0.70| 0.46 | 0.578|
> | Metric-s (GPT-O3 as judge)     | 0.70| 0.46| 0.582|
> | XCOMET-QE | 0.40| 0.29 | 0.347|
> The consistently higher performance across judges demonstrates that the observed stability is not specific to Gemini, but inherent to the design of Metric-S.
> (2) Multi-agent robustness across independent judges. To test whether prompting alone could explain the results, we applied simple, detailed, and Metric-S to Gemini:
> ### Effect of Prompting Variants (Gemini as Judge)
> | Method| System-level Consistency | Segment-level Consistency | Total Consistency |
> |-|--|--|--|
> | Metric-s + Gemini       | 0.85| 0.56| 0.703|
> | Detailed prompt + Gemini| 0.60| 0.52| 0.559|
> | Simple prompt + Gemini  | 0.20| 0.30| 0.249|
>
> The Metric-S design (structured roles, multi-dimensional scoring, error deduplication) consistently outperforms single-prompt baselines, confirming that robustness arises from the system rather than prompt artifacts.
>
> In summary, Gemini-2.5-Pro, as the judge, delivers the highest accuracy and lowest bias in judge outputs. Together with the Metric-S system, it provides an evaluation solution that achieves the highest agreement rate with human assessment, fulfilling the experimental design objectives. In the future, we will continue to explore ways to further reduce evaluation bias in large language model judges.
> # For weakness 2
> We thank the reviewer for this constructive feedback.
> First, our goal in this work is to establish a reliable evaluation framework for discourse-level and expert-level translation. As demonstrated in the paper, the metric-s system is highly reliable, showing superior correlation with human judgments, with an agreement rate of 70.3%, significantly higher than baseline XCOMET's 34.7%. Specific data can be found in Table 4 (line 358).
>
> Second, we note that several influential benchmarks in the community also adopt relatively small test suites. Although PaperBench[1] (20 tasks) and TauBench[2] (165 tasks) are modest in size, they are widely recognized benchmarks in their respective domains and serve as meaningful testbeds for validating new evaluation methodologies.
> Third, we fully agree that expanding the scale and linguistic coverage will further improve statistical robustness and generalizability. In fact, building upon this validated methodology, we are currently undertaking a major expansion of the benchmark, which includes:
> 1. Scale Enhancement: The test suite will be expanded to 500 texts, with a minimum of 50 instances per sub-category to ensure statistical power.
> 2. Linguistic Diversification: We are extending the language coverage to include Chinese-Japanese (ZH-JA), English-French (EN-FR), and English-German (EN-DE) pairs.
> In summary, the test bank will be expanded to at least 1,500 texts. This follow-up work is already in progress, and we welcome your continued interest.
>
> [1] Giulio Starace, Oliver Jaffe, Dane Sherburn, James Aung, Jun Shern Chan, Leon Maksin, Rachel Dias, Evan Mays, Benjamin Kinsella, Wyatt Thompson, Johannes Heidecke, Amelia Glaese, and Tejal Patwardhan. PaperBench: Evaluating AI's Ability to Replicate AI Research. arXiv preprint arXiv:2504.01848, 2025.
>
> [2] Shunyu Yao, Noah Shinn, Pedram Razavi, and Karthik Narasimhan. τ-bench: A Benchmark for Tool-Agent-User Interaction in Real-World Domains. arXiv preprint arXiv:2406.12045, 2024.

---

> ### Author Response · Authors · 2025-11-25
>
> # For Weakness3
> Thank you for your question.
> 1. Our Prompts Reflect Real Expert Usage (Not Under-Engineered Settings)
> The translation task prompts used in this evaluation are derived from those provided by experts during the data construction process, reflecting the actual prompts employed in expert-level and discourse-level translation scenarios， detailed in Section 2.1: Data Construction. We have retained this aspect in our evaluation to accurately assess the model’s performance in real-world user contexts rather than under heavily optimized prompt-engineering conditions.
>
> 2.  Prior Work Also Shows CoT Can Harm Translation Accuracy
> Prior work has similarly reported that explicit reasoning steps—though sometimes helpful—may introduce error propagation or unnecessary verbosity that complicates translation (e.g., ACL SRW 2025[1]). These observations are consistent with our findings.
> In addition, studies such as Mind Your Step (by Step)[2]show that CoT models may overlook explicit lexical cues, which closely mirrors the weaknesses we identify in translation tasks.
>
> 3.  Complex Prompts Improve Thinking Models—but Require Heavy Prompt Engineering
> To directly address the prompt-design concern, we conducted supplementary experiments comparing simple versus more complex, translation-specific prompts. Specifically, we sampled 50 representative cases and evaluated them using the dspk v3.1 benchmark under both prompt conditions.
> On dspk v3.1:
> - Thinking model: 43.1 (simple prompt) → 49.6 (complex prompt)
> - Non-thinking model: 45.4 (simple prompt) → 47.8 (complex prompt)
> Such gains rely heavily on carefully engineered prompts. Under simple prompts, thinking models still lag behind, and their characteristic errors (e.g., missing explicit lexical cues due to over-reasoning) persist unless the prompts are restrictive enough.
> This indicates that the issue is not merely prompt genericity; thinking models are intrinsically more sensitive to prompt formulation and require stronger behavioral constraints to perform well in translation tasks.
>
> [1] Lam Nguyen and Yang Xu. Reasoning for Translation: Comparative Analysis of Chain-of-Thought and Tree-of-Thought Prompting for LLM Translation. In Proceedings of the 63rd Annual Meeting of the Association for Computational Linguistics (Student Research Workshop), pages 259–275, Vienna, Austria, July 2025. Association for Computational Linguistics.
>
> [2] Ryan Liu, Jiayi Geng, Addison J. Wu, Ilia Sucholutsky, Tania Lombrozo, and Thomas L. Griffiths. Mind Your Step (by Step): Chain-of-Thought can Reduce Performance on Tasks where Thinking Makes Humans Worse. arXiv preprint arXiv:2410.21333, 2024.
> # For Question 1
> Helpful suggestion. In the revised version, we will integrate the following elements from the appendix into the main text:
> - The description of severity levels will be added to the Methodology section.
> - The correlation framework will be incorporated into the Experimental Setup section.
> - Additionally, ablation experiment results demonstrating the stability of Metric-S and Judge will be included in the main body.
>
> Below we provide further clarification on these components:
> 1. Severity Levels
>
> Following the MQM evaluation framework, we define four levels of error severity: minor, major, critical, and extremely critical. Each level corresponds to a specific score deduction, which is ultimately reflected in the final evaluation score.
>
> 2. Correlation Framework
>
> We adopt the consistency calculation system used in past WMT evaluations, assessing correlation at both the system level and the segment level. This framework evaluates whether the preference ordering assigned by the Judge model for different model pairs aligns with human judgments. The final correlation score is obtained by averaging the consistency rates across these dimensions.

---

### Author Response · Authors · 2025-12-03
**Summary of Contributions, Reviewers' comments and Responses to concerns**

Dear Area Chair,

Thank you for your careful work under special circumstances. We have prepared a brief summary of our work contribution, reviewer's concern and our supplementary experiments and explanations. We hope this information can help you better understand our work and what happened before.

---

## Our Contribution

Our core contribution is a comprehensive solution to evaluating long-form, professional-domain translation, comprising:

1. DiscoX Benchmark
   - The first benchmark dedicated to **discourse-level, expert-domain translation**, with **200** long-form articles from **7** professional domains (average length **>1700 tokens**), explicitly targeting discourse coherence, terminological accuracy, and stylistic consistency.

2. Metric-S Evaluation System
   - A **reference-free**, **LLM-as-judge** evaluation framework that, via a **structured multi-agent workflow**, jointly assesses **accuracy**, **fluency**, and **appropriateness**, incorporating error attribution and severity weighting, and achieving **70.3%** agreement with human experts on **DiscoX**, substantially surpassing **XCOMET-QE (34.7%)** while providing interpretable judgments.

---

## Reviewer’s Concerns

The reviews focus on three main issues:

1. **LLM‑as‑a‑judge bias and dependence**
   - Potential self‑preference when a model evaluates its own outputs.
   - Concern that our findings depend heavily on Gemini‑2.5‑Pro, both as a judge and as a system under evaluation.

2. **Dataset size and language coverage**
   - DiscoX currently has 200 items and covers EN↔ZH only, which may be viewed as limited in scale and diversity.

3. **Definitions and theoretical foundations**
   - Requests for a more precise linguistic definition of discourse.
   - Questions about the specific and detailed theoretical grounding of Metric‑S.

Below, we summarize how we address each of these concerns, including new experiments conducted after the initial rebuttal.

---

## Our Experiments and Responses

---

### A. LLM-as-a-Judge Bias and Dependence (Q1)

Our primary goal is to establish the **most effective and reliable evaluation system** for discourse-level translation. Our main experiments identified **Metric‑S with Gemini‑2.5‑Pro** as the top performer. Here we rigorously validate this choice and further prove our system's robustness.

To clarify this, we design four complementary experiments, focusing on:

1. The **factual reliability** of Gemini‑2.5‑Pro as a judge.
2. The **cross‑judge robustness** of Metric‑S under different LLM judges.
3. The impact of **Metric‑S’s internal components** via ablation.
4. The difference between **Metric‑S and single‑prompt LLM‑as‑judge** baselines using the *same* judge model.

---

### 1. Factual Reliability of Gemini‑2.5‑Pro as a Judge

**Setup.**

- Sample: **50 documents × 5 systems × 3 dimensions**
  (Accuracy, Fluency, Appropriateness).
- Procedure: Human experts review Gemini’s judgments **item by item**, checking:
  - Whether the detected error type and severity are appropriate.

**Result.**

| Evaluation Type                     | Number of Correct Cases | Total Number of Evaluations | Correct Rate |
|-------------------------------------|-------------------------|-----------------------------|--------------|
| Accurate judge           | 246                     | 250                         | 98.40%       |
| Fluency judge             | 238                     | 250                         | 95.20%       |
| Appropriateness judge       | 241                     | 250                         | 96.40%       |

**Conclusion.**

- Under our prompts and workflow, **Gemini‑2.5‑Pro is generally stable and trustworthy** as a factual judge.

---

### 2. Cross‑Judge Robustness of Metric‑S

**Setup.**

- Judge models:
  - Gemini‑2.5‑Pro
  - DeepSeek‑R1
  - GPT‑o3‑mini
- Metric compared: **Total consistency with human experts**.
- Baseline: **XCOMET‑QE (34.7%)** total consistency (reference‑free metric).

**Result.**

| Judge Type                      | System-level Consistency | Segment-level Consistency | Total Consistency |
|---------------------------------|--------------------------|---------------------------|-------------------|
| Metric-s (Gemini as judge)      | 85.00%                   | 56.00%                    | 70.30%            |
| Metric-s (DeepSeek-R1 as judge) | 70.00%                   | 46.00%                    | 57.80%            |
| Metric-s (GPT-O3 as judge)      | 70.00%                   | 46.00%                    | 58.20%            |
| XCOMET-QE                       | 40.00%                   | 29.00%                    | 34.70%            |

**Conclusion.**

- The results confirm that Gemini-2.5-Pro is the **best judge** (70.3% consistency). But more importantly, our Metric-S framework significantly **outperforms** the baseline across all judges, proving its performance advantage comes from the **framework** design itself, not the capability of a specific LLM.

---

> ### Author Response · Authors · 2025-12-03
>
> ### 3. Ablation of Metric‑S Components
>
> **Goal.**
> Understand which parts of Metric‑S contribute most to the performance gains.
>
> **Key components examined.**
>
> 1. **Three‑dimensional evaluation**
>    (Accuracy, Fluency, Appropriateness).
> 2. **Hierarchical de‑duplication**
>    (merging repeated or overlapping errors across spans and dimensions).
>
> **Setup.**
>
> - Fix the judge model as **Gemini‑2.5‑Pro**.
> - Compare total consistency with humans under different variants.
>
> **Result.**
>
> | Judge Type                                  | Accuracy | Fluency | Appropriateness | De-duplication | Consistency |
> |---------------------------------------------|----------|---------|-----------------|----------------|-------------|
> | Metric-S (Gemini as judge)                  | √        | √       | √               | √              | 70.30%      |
> | Metric-S (Gemini as judge + No-deduplicate) | √        | √       | √               | ×              | 66.00%      |
> | Metric-S (Gemini as judge + Accuracy only)  | √        | ×       | ×               | ×              | 62.50%      |
>
> **Conclusion.**
>
> - **Accuracy** is the dominant driver of scores aligned with human judgments.
> - For **discourse-level evaluation**, **Fluency** and **Appropriateness** are indispensable: they capture global coherence, stylistic consistency, and register.
> - The **hierarchical de‑duplication mechanism** further improves reliability by reducing noise from repeated/overlapping errors.
> - Together, these components yield a **+7.8%** gain over an Accuracy‑only variant.
>
> ---
>
> ### 4. Metric‑S vs. Single‑Prompt LLM‑as‑Judge
>
> **Goal.**
> Is the improvement due to Metric‑S’s **multi‑agent, staged, rubric‑guided design**, or simply because we use a strong LLM?
>
> **Setup.**
>
> - Same judge model: **Gemini‑2.5‑Pro**.
> - Compare:
>   1. **Metric‑S + Gemini** (our full framework)
>   2. **Detailed‑prompt + Gemini**
>      - Single‑prompt LLM‑as‑judge baseline with a rich, explicit rubric.
>   3. **Simple‑prompt + Gemini**
>      - Minimal, generic rating prompt.
>
> **Result.**
>
> | Method                   | System-level Consistency | Segment-level Consistency | Total Consistency |
> |--------------------------|--------------------------|---------------------------|-------------------|
> | Metric-s + Gemini        | 85.00%                   | 56.00%                    | 70.30%            |
> | Detailed prompt + Gemini | 60.00%                   | 52.00%                    | 55.90%            |
> | Simple prompt + Gemini   | 20.00%                   | 30.00%                    | 24.90%            |
>
> **Conclusion.**
>
> - With the **same judge model**, Metric‑S improves total consistency by:
>   - **+14.4%** over a strong detailed‑prompt baseline (55.9% → 70.3%),
>   - **+45.4%** over a simple‑prompt baseline.
> - This shows that the **Metric‑S framework itself**—especially its staged evaluation, multi‑dimensional rubric, and hierarchical de‑duplication—is the main driver of performance, not just the power of the underlying LLM.
>
> ---
>
> In summary, these supplementary experiments, combined with our main results, conclusively demonstrate that **the combination of Metric‑S and Gemini‑2.5‑Pro constitutes the best and most reliable evaluation system** for our task.
>
> ---
>
> ### B. Dataset Size and Language Coverage (Q2)
>
> Our main contribution is a **reliable evaluation framework** for discourse‑level, expert‑domain translation. On DiscoX, Metric‑S already achieves **70.3%** total consistency with human experts, showing that the current **200 long, expert‑curated documents** are sufficient to support our conclusions.
>
> We fully agree that **larger scale and broader language coverage** are important. This is the focus of our ongoing work:
>
> - Expanding to **≈500** EN–ZH documents, with ≥50 items per sub‑domain.
> - Extending to **≥1,500** documents across multiple language pairs (e.g., **ZH–JA**, **EN–FR**, **EN–DE**) using the same construction pipeline and Metric‑S framework.
>
> ---

---

> > ### Author Response · Authors · 2025-12-03
> >
> > ### C. Definitions and Theoretical Foundations (Q3)
> >
> > ---
> >
> > ### 1. Discourse: Linguistic Definition and Our Usage
> >
> > Our goal is not to redefine “discourse” linguistically, but to **evaluate translation** based on established discourse theory.
> >
> > In mainstream linguistics, **discourse** is language **beyond the sentence**, characterized by:
> >
> > - **(i) Supra‑sentential structure**
> >   Multiple sentences/paragraphs form a coherent event, argument, or narrative.
> >
> > - **(ii) Context‑dependence**
> >   Interpretation depends on physical, cultural, pragmatic, and interpersonal context.
> >
> > - **(iii) Cohesion and coherence**
> >   Cohesion via pronouns, ellipsis, connectives, lexical repetition;
> >   Coherence via deeper logical and inferential relations between discourse segments.
> >
> > **How DiscoX operationalizes discourse**
> >
> > DiscoX encodes these properties via:
> >
> > 1. **Text choice**
> >    - 200 long, real‑world professional documents (avg. **1,712 tokens**).
> >    - 7 expert domains (law, medicine, biomedicine, science & technology, social sciences, etc.).
> >    - Naturally rich in cross‑sentence references, terminological chains, and global argument structures.
> >
> > 2. **Expert rubrics**
> >    - For each document, experts annotate **document‑specific rubrics** that highlight:
> >      - Cross‑sentence/coreference dependencies.
> >      - Terminology and conceptual chains that must remain consistent.
> >      - Genre conventions and culturally loaded expressions.
> >
> > These rubrics translate abstract notions like **coherence, terminological chains, genre/register, pragmatics** into **concrete, document‑specific evaluation criteria**.
> >
> > ---
> >
> > ### 2. Metric‑S: Theoretical Foundation
> >
> > Our evaluation dimensions are **not ad‑hoc**; they are a principled consolidation of the **Multidimensional Quality Metrics (MQM)** framework, the most widely used theoretical basis for translation quality assessment.
> >
> > **Motivation**
> >
> > - Recent discourse‑level work, especially the **WMT 2024 Literary Task**, shows that standard metrics (BLEU, COMET) **critically fail** on document‑level translation, with human–metric system ranking agreement often below **53%**.
> > - This exposes an urgent need for a **more reliable evaluation system** for discourse‑level, expert‑domain translation—precisely what Metric‑S is designed for.
> >
> > **Mapping MQM → Metric‑S dimensions**
> >
> > Instead of directly applying MQM's 20+ fine‑grained categories—which are impractical for LLM‑based evaluation of long texts and cause issues like inconsistent judgments and error inflation—we consolidate them into three robust, MQM‑aligned macro‑dimensions:
> >
> > | MQM Domain                               | Metric-S Dimension  | Rationale                                                                                      |
> > |------------------------------------------|---------------------|------------------------------------------------------------------------------------------------|
> > | **Accuracy branch** (Mistranslation, Omission, Addition, Terminology, Factual errors) | **Accuracy**        | Fully aligned; DiscoX rubrics (Appendix B.3) fall squarely in this branch.                    |
> > | **Fluency branch** (Grammar, Clarity, Coherence) | **Fluency**   | Matches MQM's fluency & coherence categories; essential for long, multi-paragraph texts.      |
> > | **Style & Pragmatics branch** (Register, Style, Tone, Cultural appropriateness) | **Appropriateness** | Consolidation of stylistic and pragmatic MQM categories frequently seen in expert-domain texts. |
> >
> > This consolidation is **not an arbitrary simplification**, but a **methodological necessity** for evaluating complex, long-form texts with LLM judges. It preserves MQM’s theoretical coverage while ensuring:
> >
> > - **Practical usability** for LLM‑as‑judge on long documents.
> > - **Reduced error inflation** via our hierarchical de‑duplication.
> > - **High agreement** with human experts, as demonstrated in our experiments.
> >
> > ---
> >
> > That's all for this brief. We welcome you to read the full edition of our paper and comments. Thank you again for your time.

---

### Meta-Review · Area_Chair_ssVk · 2026-01-06

**Summary:**

This paper introduces DiscoX, a new benchmark for discourse-level, expert-domain machine translation, and Metric-S, a corresponding reference-free evaluation framework.

The reviewers' concerns that informed the decision revolved around three main issues: (1) The reliability of the LLM-as-judge evaluation, specifically its dependence on a single model (Gemini-2.5-Pro) and potential for bias. (2) The theoretical foundations of the work, with one reviewer arguing that key concepts like ``discourse-level translation'' were ill-defined and that the Metric-S evaluation was ad hoc. (3) The limited scale of the benchmark (200 texts, EN↔ZH only) and various points of unclarity in the paper's presentation and methodology.

**Reviewer Concerns:**

Addressed by the rebuttal and discussion-equivalent materials:
- **LLM-as-judge reliability and dependence on Gemini-2.5-Pro (concerns from Lndd, y79T):** This was fully addressed. The authors conducted multiple new experiments demonstrating (a) high factual reliability of the Gemini judge (>95% correctness via human verification), (b) robustness of the Metric-S framework, which significantly outperforms baselines even with different judge models (DeepSeek-R1, GPT-o3-mini), and (c) the value of the framework's design itself, which provides a +14.4% gain over a strong single-prompt baseline using the same judge. This shows the contribution is the framework, not just the use of a powerful model.
- **Theoretical grounding and definitions (concerns from A2t8, DyVE):** This was fully addressed. The authors provided a clear linguistic definition of ``discourse'' grounded in established literature (e.g., cohesion, coherence, context-dependence) and demonstrated that Metric-S's dimensions are a principled consolidation of the widely accepted Multidimensional Quality Metrics (MQM) framework.
- **Weighting justification and methodological clarity (concerns from y79T, DyVE):** This was fully addressed. The authors provided a sensitivity analysis showing their 60/20/20 weighting for Accuracy/Fluency/Appropriateness better aligns with human judgment than uniform weighting. They also answered all specific clarification questions on terminology (e.g., "tokens", "expert-level") and committed to integrating these details into the final paper.

Partially addressed or outstanding as limitations to be framed in the final paper:
- **Dataset scale and language coverage (concerns from Lndd, A2t8, y79T):** This is an inherent limitation of the current work. The authors acknowledged this, justified the current scale as sufficient for validating their methodology, and outlined concrete plans for future expansion. This is an acceptable limitation for a paper introducing a new resource and methodology, but claims should be scoped accordingly.
- **Presentation and Organization:** Key methodological details (e.g., severity levels, MQM mapping) must be moved from the rebuttal and appendices into the main text for the camera-ready version to improve self-containedness.


Ethics:
- The forum indicates the AC triage decision is ``Do NOT proceed with ethics review''. I agree with this assessment. However, given reviewer concerns about legal compliance, the final paper would benefit from a concise statement on data provenance and redistribution rights.

**Reviewer Scores:**

- **Reviewer Lndd (original: 6):** likely remain at 6. The primary concern about judge dependence was convincingly addressed by new experiments.
- **Reviewer A2t8 (original: 2):** likely increase to 4. The reviewer's core criticisms about lack of theoretical grounding were directly refuted by the authors' mapping to MQM and linguistic literature. While the reviewer might still have concerns about presentation or scale, the substantive reasons for the strong reject are resolved.
- **Reviewer DyVE (original: 6):** likely remain at 6. Most issues were requests for clarification, all of which the authors answered thoroughly.
- **Reviewer y79T (original: 6):** likely remain at 6. Requests for robustness tests and weighting justification were directly met with new experiments.

---

### Decision · Program_Chairs · 2026-01-26

Accept (Poster)